# MYB repressors and MBW activation complex collaborate to fine-tune flower coloration in *Freesia hybrida*

Yueqing Li [1,4], Xiaotong Shan[1,4], Ruifang Gao[1], Taotao Han[1], Jia Zhang[1], Yanan Wang[1], Shadrack Kimani[1,2], Li Wang [1✉] & Xiang Gao [1,3✉]

Floral anthocyanin has multiple ecological and economic values, its biosynthesis largely depends on the conserved MYB-bHLH-WD40 (MBW) activation complex and MYB repressors hierarchically with the MBW complex. In contrast to eudicots, the MBW regulatory network model has not been addressed in monocots because of the lack of a suitable system, as grass plants exhibit monotonous floral pigmentation patterns. Presently, the MBW regulatory network was investigated in a non-grass monocot plant, *Freesia hybrida*. FhMYB27 and FhMYBx with different functional manners were confirmed to be anthocyanin related R2R3 and R3 MYB repressors, respectively. Particularly, FhMYBx could obstruct the formation of positive MBW complex by titrating bHLH proteins, whereas FhMYB27 mainly defected the activator complex into suppressor via its repression domains in C-terminus. Furthermore, the hierarchical and feedback regulatory loop was verified, indicating the synergistic and sophisticated regulatory network underlying *Freesia* anthocyanin biosynthesis was quite similar to that reported in eudicot plants.

[1] Key Laboratory of Molecular Epigenetics of MOE and Institute of Genetics & Cytology, Northeast Normal University, Changchun, China. [2] School of Pure and Applied Sciences, Karatina University, Karatina, Kenya. [3] National Demonstration Center for Experimental Biology Education, Northeast Normal University, Changchun, China. [5]These authors contributed equally: Yueqing Li, Xiaotong Shan ✉email: wanglee57@163.com; gaoxiang424@163.com

About 450 million years ago, sunlight, carbon dioxide and mineral nutrients appealed to the terrestrial plant predecessors. Meanwhile, the challenges of intensive ultra violet rays, deficient water and their uncomplicated bodies drove the evolutionary emergence of entirely new specialized metabolites, such as flavonoids and lignins[1–3]. Flavonoids shield plants from ultra violet rays and are involved in transporting auxin, attracting pollinators or seed dispersers[2,4–6], they are ubiquitously distributed in the plant kingdom and can be grouped into flavones, flavonols, flavanones, proanthocyanidins (PAs) and anthocyanins. Additionally, the flavonoids metabolic pathways are conserved, although subclasses seem to emerge at different times during evolution. For example, flavones have already been biosynthesized in liverwort[2]. Comparably, PAs and anthocyanins are widely distributed in relatively higher plants[7].

Generally, the biosynthesis and accumulation of specific flavonoids in plant tissues and developmental stages are fine-tuned and tightly regulated by transcription factors. In all studied species, the expression of anthocyanin structural genes is conservatively controlled by the canonical MBW complex[4,8]. Additionally, the MBW complexes take part in PA biosynthesis, cell fate determination, vacuolar acidification adjustment[4,8]. The highly conserved WD40 proteins have been hypothesized to provide a scaffolding role in boosting gene transcription driven by the MYB-bHLH complex. The bHLH factors can be divided into bHLH1 (including *Arabidopsis thaliana* GL3/EGL3, petunia JAF13, maize R, etc.) and bHLH2 (including *Arabidopsis* TT8, petunia AN1, maize IN1, etc.) clades[9,10]. Comparatively, the MYB components have been characterized to be the most specific and conspicuous regulators determining anthocyanin patterns. Until now, MBW components have been widely investigated in eudicot plants, such as petunia, *Arabidopsis*, *Mimulus*, grapevine, apple and peach[5,11–16]. Recently, in addition to maize[5], a growing number of MBW factors have also been characterized in monocots including lily, orchid, onion and rice[17–21]. However, few studies delved into the mutual regulation among the MBW components except in model *Arabidopsis*, *Mimulus*, petunia and tobacco[9,10,15,22].

Beyond the positive MBW complex aforementioned, MYB repressors have also been investigated in recent years. Generally, two types of MYB repressors participate in anthocyanin regulation, R2R3-MYB with R2 and R3 repeats and R3-MYB with only R3 repeat, respectively. R2R3-MYB repressors represented by petunia MYB27 and grapevine MYBC2s usually possess characteristic repression domains, while R3-MYB repressors such as *Arabidopsis* CAPRICE (CPC) and petunia MYBx lack repression motifs in the C-termini[10,23]. An exception is *Arabidopsis* MYBL2, whose R2 domain is largely missing because of the truncation of its first exon, it is phylogenetically and functionally related to the R2R3-MYB repressors, however[24]. Since the elegant work with petunia MYB27 and MYBx illuminated the network of anthocyanin related transcription activators and repressors, increasing MYB repressors have been characterized in model and crop dicotyledons to negatively regulate anthocyanin biosynthesis in relatively conserved ways[10,15,23,25–32]. Previous analysis of the evolutionary rates of the anthocyanin biosynthetic genes (ABGs) indicated that the late biosynthetic genes (LBGs, including *DFR*, *LDOX* and *3GT*) evolved more quickly than the early biosynthetic genes (EBGs, including *CHS*, *CHI* and *F3H*), which may be resulted from the profound influence of EBGs on diverse flavonoids[33]. Comparatively, the regulatory genes in anthocyanin biosynthesis may be less evolutionarily constrained as they evolved more rapidly than the structural genes[33–35]. Herein, whether the hierarchical and feedback mechanisms underlying anthocyanin biosynthesis are conserved in monocotyledons contrariwise to eudicot plants remains largely unstudied, though

MYB repressors have been sporadically reported in monocot plants[36–38]. Moreover, the MBW regulatory network responsible for floral anthocyanin biosynthesis in monocots has not been addressed to our knowledge, which needs particularly attentions, because floral pigmentation is crucial for signaling to insect and animal pollinators and is also an important trait to determine the ornamental values of horticultural plants[34].

Plants from *Iridaceae* family especially *Freesia* species are best-known for the colorful large flowers and impressive pigmentation types. Previous studies revealed that flowers of different *F. hybrida* cultivars accumulated abundant flavonoids including anthocyanins, PAs and flavonols[39–41]. Additionally, flower anthocyanin accumulation patterns were tightly regulated in a spatio-temporal manner, making *F. hybrida*, a representative and powerful system to explore chemical ecology, genetics and regulation of anthocyanin biosynthesis. In *Freesia* flowers, five anthocyanin aglycons (delphinidin, cyanidin, petunidin, peonidin and malvinidin) and the ABGs have been investigated in the widely cultivated red flowered cultivar Red River®[40–46]. Moreover, the versatile transient expression system based on the flower petals or protoplasts isolated from *Freesia* callus partially overcomes the obstacle of the recalcitrant characters of monocots against *Agrobacterium* and immensely accelerates the molecular investigations in *Freesia per se*[47,48]. To date, the components of MBW activator complex have been elucidated in *Freesia* and the regulatory characteristics are conserved but with slight differences compared with their orthologs in model plants[39,48–50]. However, the hierarchical and feedback gene regulatory network of *Freesia* anthocyanin biosynthesis including MYB repressors remains unresolved.

In this study, FhMYB27 and FhMYBx, belonged to R2R3-MYB and R3-MYB subgroup respectively, were isolated from *Freesia* flowers and functionally characterized. Functional studies indicated that they could suppress anthocyanin biosynthesis by inhibiting ABGs expression and might function in differential mechanisms according to repressor domains in the C-terminus. The fact that *FhMYB27* and *FhMYBx* could be activated by the MYB activator FhPAP1 prompted a regulatory loop to fine-tune the anthocyanin accumulation in *Freesia*. Our study attempts to reveal the hierarchical and feedback control of *Freesia* anthocyanin biosynthesis towards elaboration of the anthocyanin regulatory mechanisms in plants at different evolutionary positions.

## Results

**FhMYB27 and FhMYBx encode different MYB repressors.** After *in situ* TBLASTN screen of *Freesia* transcriptomic database and sequence analysis by manual NCBI-BLASTX search, two genes encoding orthologs of *PhMYB27* and *PhMYBx* were mined and displayed high similarities to MYB repressors in other plants, and thus designated as *FhMYB27* and *FhMYBx*, respectively. The *FhMYB27* gene encoded a predicted R2R3-MYB protein with 208 amino acid residues, while *FhMYBx*, a potential R3-MYB gene had only 74 amino acid residues (Supplementary Table 1). Subsequently, the two MYB proteins were aligned with counterparts in other plants. Both FhMYB27 and FhMYBx had the [D/E]Lx$_2$[R/K]x$_3$Lx$_6$Lx$_3$R motif (Fig. 1a), a central feature for proteins that function by interacting with bHLH factors. Moreover, FhMYB27 C-terminus contained additional conserved motifs, such as C1-like motif (LlsrGIDPxT/SHRxI/L), C2-like motif (pdLNLD/ELxiG/S), C5-like repressor motif (TLLLFR), and the LxLxL-type EAR (ethylene response factor-associated amphiphilic repression) repressor domain[24,28,51]. Notably, the C1-like motif and the C2-like motif are main hallmarks of subgroup 4 MYB proteins. When matched to the lignin or phenylpropanoid repressors such as AtMYB4 and EgMYB1, FhMYB27 lacked the

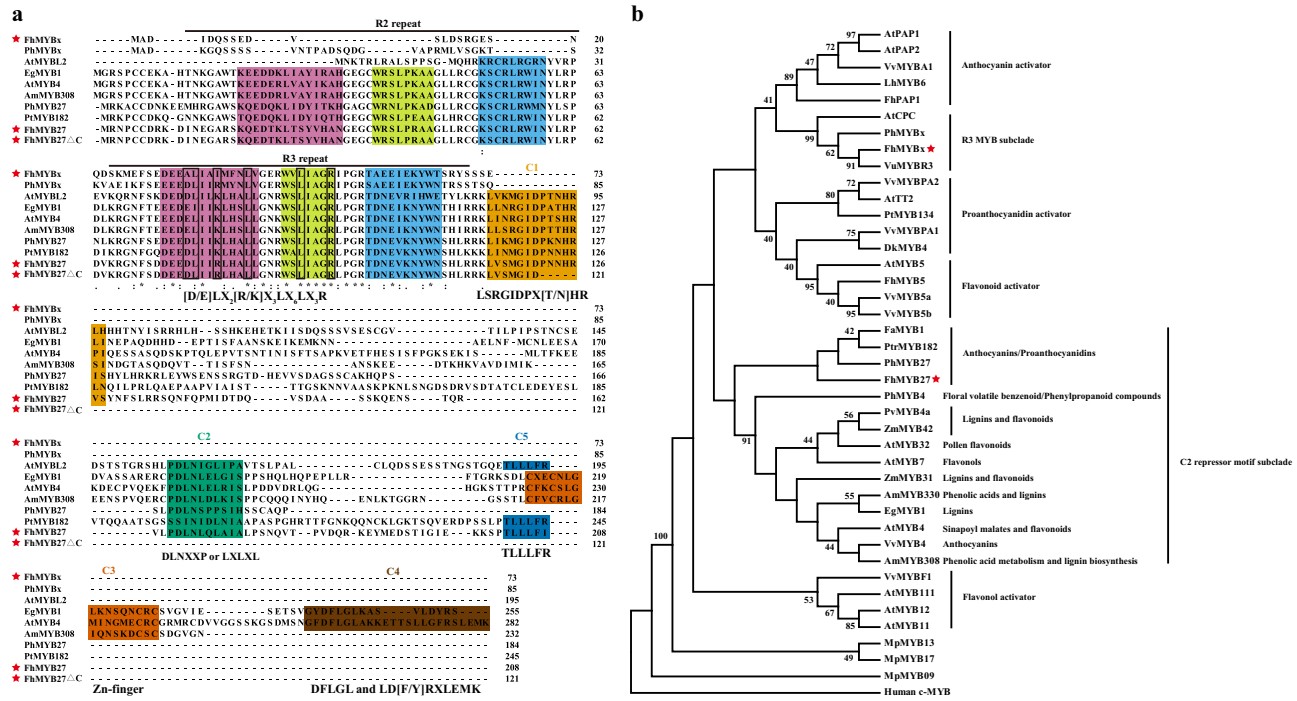

**Fig. 1 Amino acid sequence alignment and phylogenetic analysis of FhMYB27, FhMYBx, and other MYB proteins. a** Multiple alignment of the amino acid sequences of anthocyanin or proanthocyanidin related MYB repressors. Modified version of FhMYB27 was included: FhMYB27ΔC (C-terminal repression domain removed). Conserved residues were indicated by asterisk and partial conservations were indicated by: or ·. The numbers indicated the positions of the last residue in each line. The R2 and R3 domains were indicated by overlines. The shaded residues in R2 and R3 domains indicated the three helixes with different colors, respectively. The conserved [D/E]Lx2[R/K]x3Lx6Lx3R motif responsible for interacting with bHLH factors were boxed with rectangular frames. The conserved C1, C2, C3, C4, and C5 motifs in the C-terminus of different MYB repressors were outlined with different colors. Red stars indicated the *Freesia* MYB regulators isolated in this study. **b** Phylogenetic analysis of amino acid sequences of MYB proteins in *Freesia* and other species. The *Freesia* FhMYB27 and FhMYBx were highlighted with the red stars. The tree was constructed using Maximum Likelihood method and Poisson correction model by MEGA version X. The human c-MYB sequence (NP_001155129) was used for the outgroup, and a selection of non-flavonoid-related R2R3MYBs from *Marchantia polympha* (MpMYB09 PTQ41991.1; MpMYB13 PTQ35332.1; MpMYB17 PTQ32714.1) were included for comparison. Nodes with bootstrap no <40% from 1000 replicates were shown. The following GenBank accession numbers were used: *Antirrhinum majus* AmMYB308 (P81393), AmMYB330 (P81395); *Arabidopsis thaliana* AtPAP1 (AAG42001), AtPAP2 (AAG42002), AtTT2 (Q2FJA2), AtMYB11 (NP_191820), AtMYB12 (CAB09172), AtMYB111 (NP_199744), AtMYB4 (AAC83582), AtMYB7 (NP_179263), AtMYB32 (NP_195225), AtMYB5 (AAC49311), AtCPC (NP_182164); *Vitis vinifera* VvMYBF1 (ACT88298), VvMYB5a (AAS68190), VvMYB5b (Q58QD0), VvMYBPA1 (CAJ90831), VvMYBPA2 (ACK56131), VvMYB4 (NP_001268129); *Petunia hybrida* PhMYB4 (ADX33331), PhMYB27 (AHX24372), PhMYBx (AHX24371); *Lilium hybrid* LhMYB6 (BAJ05399); *Zea mays* MYB31 (NP_001105949), ZmMYB42 (NP_001106009); *Diospyros kaki* DkMYB4 (BAI49721); *Fragaria ananassa* FaMYB1 (AAK84064); *Populus tremula x Populus tremuloides* PtrMYB182 (AJI76863), PtrMYB134 (ACR83705); *Eucalyptus gunnii* EgMYB1 (CAE09058); *Panicum virgatum* PvMYB4a (AEM17348); and *Vaccinium uliginosum* VuMYBR3 (AKR80572).

C3 zinc-finger motif (CX1-2CX7-12CX2C) and C4 motif (FLGLx4-7V/LLD/GF/YR/Sx1LEMK)[28,52].

To better define FhMYB27 and FhMYBx, phylogenetic analysis with other repressor and activator MYB regulators was evaluated. The phylogeny in Fig. 1b implied that a number of clades were resolved implicating their different functions. Expectedly, anthocyanin, flavonol and PA biosynthesis related activators were defined with authentic bootstrap values. The new *Freesia* FhMYB27 grouped with the previously characterized anthocyanin or PA repressors petunia PhMYB27 and poplar PtrMYB182, while FhMYBx clustered with R3 MYB subclade responsible for specialized metabolism or cell fate determination. Together with the protein alignment analysis in Fig. 1a, the results indicated FhMYB27 and FhMYBx might be negative regulators involved in anthocyanin or proanthocyanidin biosynthesis.

In order to determine the subcellular localization of FhMYB27 and FhMYBx, the two proteins were fused with GFP to construct *35 S: FhMYB27/FhMYBx-GFP* or *35 S: GFP-FhMYB27/FhMYBx* vectors which were transiently transformed into *Arabidopsis* protoplasts. The nuclear signals of GFP were captured in protoplasts overexpressing *FhMYB27-GFP* or *GFP-FhMYB27*,

whereas GFP alone, FhMYBx-GFP and GFP-FhMYBx showed both cytoplasmatic and nuclear signals (Supplementary Fig. 1). Hence, FhMYB27 and FhMYBx might function as transcriptional regulators in plant.

**The spatiotemporal expression patterns of *FhMYB27* and *FhMYBx*.** Our previous work proved that the anthocyanins and PAs had different accumulating patterns in *F. hybrida*[39,49]. The anthocyanins increased throughout flower development, whereas PAs fluctuated at a relatively stable level toward blooming. As for the floral tissues or organs, anthocyanins and PAs were most sufficiently accumulated in petals and toruses, respectively. In order to get putative relationships between the *FhMYB27* and *FhMYBx* factors and anthocyanins or PAs, the spatiotemporal expression patterns were analyzed in flower development process as well as vegetative and reproductive tissues or organs (Fig. 2a). The constructed heat map allowed visualization of the general gene expression tendencies of *FhMYB27* and *FhMYBx*, as well as *Freesia* anthocyanin biosynthetic genes and the earlier characterized MBW components[39,48,50]. The *FhMYB27* and *FhMYBx*

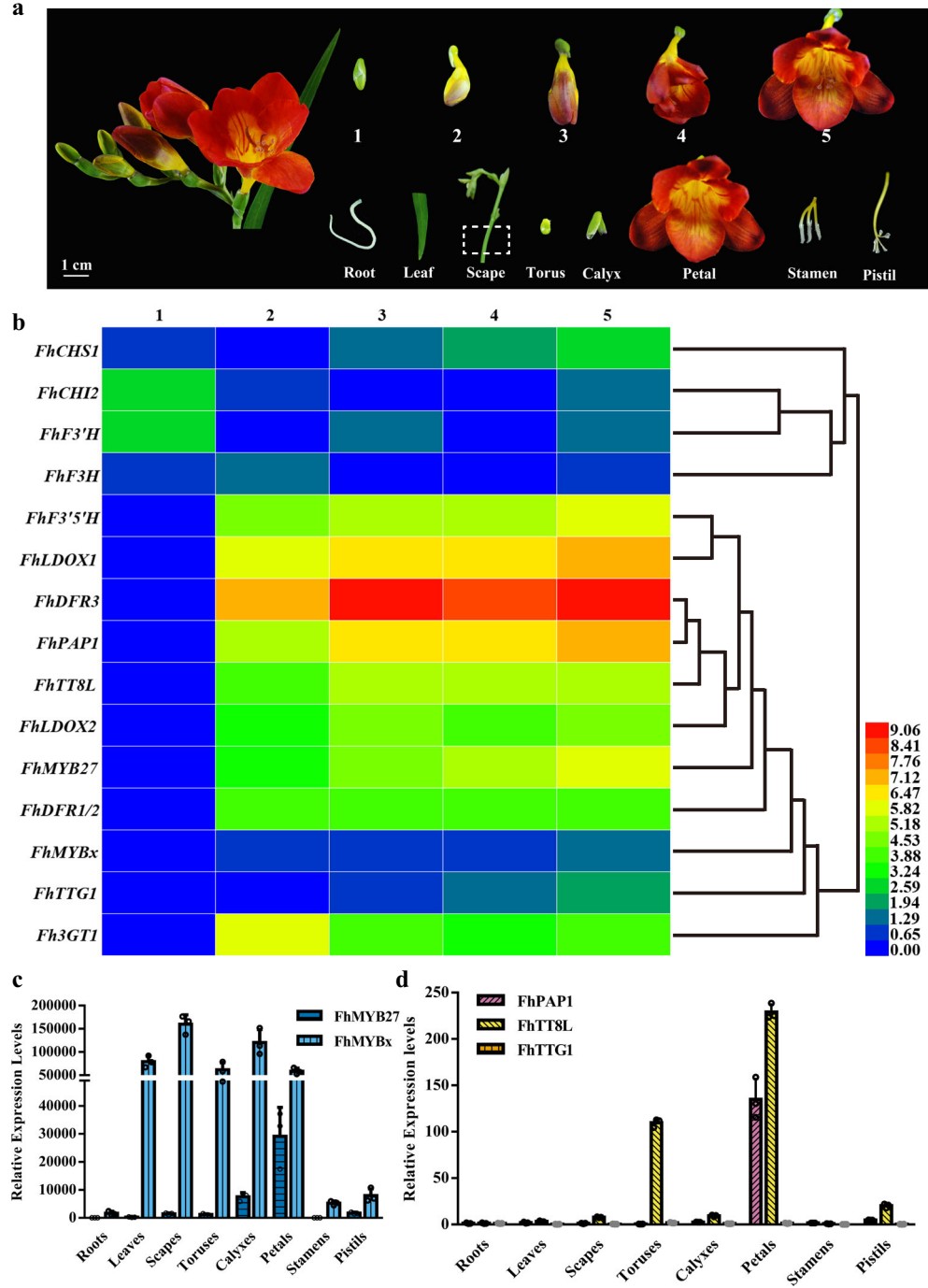

**Fig. 2 Expression profiles of anthocyanin biosynthesis related genes in *F. hybrida*. a** The phenotypic trait, flower developmental stages and different tissues of *F. hybrida* cultivar Red River®. The numbers indicated the five developmental stages. Stage 1, <10 mm long with unpigmented buds; Stage 2, 10–20 mm long with slightly pigmented buds; Stage 3, 20–30 mm long with pigmented buds; Stage 4, fully pigmented flowers before completely opening; Stage 5, fully opened flowers. The box with dotted lines represented the segment sampled as scape. **b** Expression analysis of anthocyanin biosynthetic genes by qRT-PCR at different developmental stages of *Freesia* flowers. 1–5, represented the flowers at different developmental stages. Data represented changes relative to the least expression level of each gene in specific developmental stage. The mean data of three replicates were calculated as log$_2$, and were hierarchically clustered based on average Pearson's distance metric. Red and blue boxes indicated high and low expression levels respectively. **c** Expression profiles of *FhMYB27* and *FhMYBx* in different tissues. Data represented changes relative to the least expression level of *FhMYB27*. **d** Expression profiles of the potential MBW components in different tissues. Data represented changes relative to the least expression level of each gene in root. Data indicated the mean ± SD of three biological replicates in **c** and **d**.

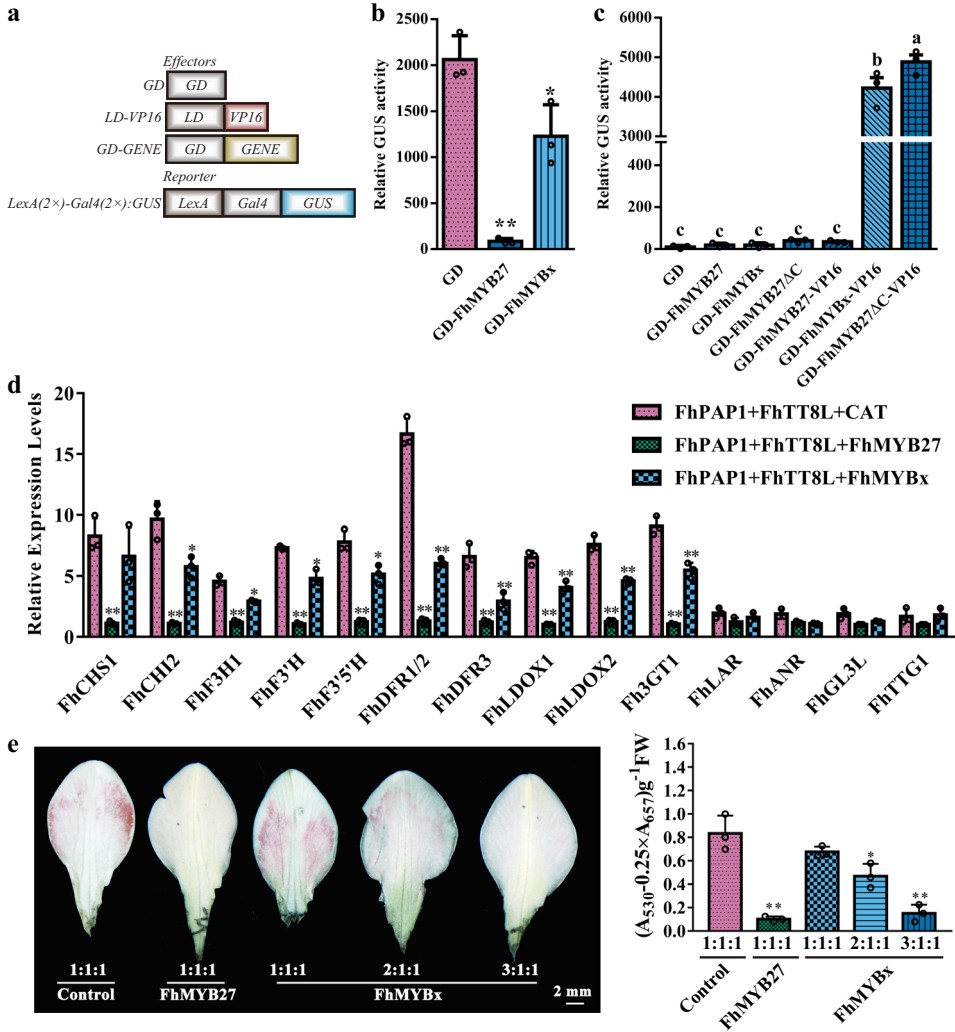

**Fig. 3 FhMYB27 and FhMYBx were anthocyanin biosynthesis related repressors in *F. hybrida*. a** The schematic diagram of LexA-Gal4: GUS system. The reporter plasmid LexA-Gal4: GUS contained LexA DNA binding site, Gal4 DNA binding site and GUS reporter gene. GUS could be activated by LD-VP16 containing LexA-DNA binding domain (LD) fused VP16. GD (*Saccharomyces cerevisiae* Gal4 DNA binding domain) fused target gene could bind to the Gal4 DNA binding site. The GUS activity activated by LD-VP16 could be reduced to different levels depending on the transrepression capacity of the GD-fused protein. As negative, the effector containing only GD had no effect on GUS activity. **b** The transcriptional repression capacities of FhMYB27 and FhMYBx detected by LexA-Gal4: GUS system. **c** The transactivation capacities of FhMYB27 and FhMYBx detected by Gal4: GUS system. The reporter plasmid Gal4: GUS contained the Gal4 DNA binding site and GUS reporter gene. The effector plasmid contained GD-fused target gene. The GUS would be activated when the GD-fused protein was an activator. Modified versions of MYB repressors were also assayed: FhMYB27ΔC (C-terminal repression domain removed), FhMYB27-VP16 (viral activation domain added), FhMYB27ΔC-VP16 (C-terminal repression domain removed + viral activation domain), FhMYBx-VP16 (viral activation domain added). **d** Expression of anthocyanin biosynthetic genes could be repressed by FhMYB27 or FhMYBx in transgenic *Freesia* protoplasts. *FhMYB27*, *FhMYBx* or the irrelevant *CAT* was cotransfected with *FhPAP1* and *FhTT8* into *Freesia* protoplasts. After 21 hours incubation, total RNA was extracted and gene expression levels were checked by qRT-PCR. **e** The phenotypes and relative anthocyanin contents of Ambiance petals transiently expressing different regulators. The *Agrobacterium tumefacien* GV3101 containing empty vector, *FhMYB27* or *FhMYBx* was mixed with those expressing *FhPAP1* and *FhTT8L* at different ratios and injected into Ambiance buds. Numbers indicated the ratios between empty vector, *FhMYB27* or *FhMYBx* and *FhPAP1* and *FhTT8L*. Data represented the mean ± SD of three biological replicates. Student's *t*-test was computed to analyze the significant difference in **b**, **d** and **e** (*$p < 0.05$; **$p < 0.01$). One-way ANOVA was carried out to compare statistical differences in **c** (Duncan's test, $p < 0.05$).

were grouped together with LBGs and MBW components, which were shown to concurrently increase during flower development (Fig. 2b). The data derived from different tissues or organs also underlined the potential roles of the two MYB factors in anthocyanin biosynthesis considering their high transcripts in colored petals (Fig. 2c). Comparatively, *FhMYBx* was substantially expressed in non-colored *Freesia* tissues or organs and had highest expression in all the tests, indicating FhMYBx and FhMYB27 might function differently in *F. hybrida*. Moreover, the

*Freesia* MBW components, *FhPAP1*, *FhTT8L* and *FhTTG1*, also showed highest expression levels in petals (Fig. 2d), implying possible correlations between FhMYB27 or FhMYBx and the MBW complex in floral anthocyanin biosynthesis.

**FhMYB27 and FhMYBx repress anthocyanin biosynthesis**. The sequence alignment, phylogenetic and expression analysis conceivably implied that FhMYB27 and FhMYBx were anthocyanin related repressors. Herein, the widely accepted LexA-Gal4: GUS

was employed in analysis (Fig. 3a)[53]. Briefly, the reporter vector LexA-Gal4: GUS was cotransfected into protoplasts with the effector plasmid LD-VP16. The LD-VP16 could bind to *LexA* and then activate *GUS* expression by the strong activation domain VP16[54]. Simultaneously, GD-tagged *FhMYB27* or *FhMYBx* was cotransfected with LexA-Gal4: GUS and LD-VP16. If FhMYB27 or FhMYBx was transcription repressor, it was supposed to repress *GUS* expression activated by VP16 as the GD tag could specially bind to the *Gal4* sequence. The FhMYB27 and FhMYBx remarkably reduced *GUS* expression levels and FhMYB27 showed stronger transrepression capacity than FhMYBx, corroborating their roles as repressors (Fig. 3b).

Consistently, data derived from the formerly used Gal4: GUS system showed neither GD-tagged FhMYB27 nor GD-tagged FhMYBx had transactivation capacity (Fig. 3c). To further elucidate the effects of different domains on the transcriptional activity of the repressors, the strong activation domain VP16 was fused to the C-terminus of FhMYB27 or FhMYBx. Consequently, VP16 caused the conversion of FhMYBx from repressor into activator. However, FhMYB27 seemed to be resistant considering the existing repression domains in its C-terminus (Fig. 3c). Subsequently, FhMYB27ΔC and FhMYB27ΔC-VP16 were formed by removing the C-terminal repression domains solely or substituting with the viral activation domain VP16. Only GD-tagged FhMYB27ΔC-VP16 prominently activated *GUS* expression (Fig. 3c).

As substantiated in our earlier experiments, the *Freesia* FhPAP1 and FhTT8L could form MYB-bHLH complex to activate the ABGs expression[39,48]. To test the prediction that FhMYB27 and FhMYBx were anthocyanin related repressors, they were cotransfected with *FhPAP1* and *FhTT8L* into *Freesia* protoplasts isolated from the callus of Red River® cultivar. The expression analysis of anthocyanin structural genes examined by qRT-PCR revealed that FhMYB27 significantly repressed ABGs expression activated by FhPAP1 and FhTT8L. Analogously, FhMYBx also yielded significant but partial repression activities on nearly all ABGs expression (Fig. 3d). To test the prediction that FhMYB27 or FhMYBx could finally reduce anthocyanin levels by down-regulating the anthocyanin pathway, Ambiance petals overexpressing *FhPAP1* and *FhTT8L* were transiently infected by *Agrobacterium* containing *FhMYB27* or *FhMYBx*. As results showed in Fig. 3e, FhMYB27 could efficiently repress anthocyanin biosynthesis activated by FhPAP1 and FhTT8L, whereas the repressing effect of FhMYBx on anthocyanin biosynthesis largely depended on its dosage. Correspondingly, the total anthocyanins in different transgenic Ambiance petals corroborated the expression of *FhMYB27* and *FhMYBx*.

To understand their roles in whole plants, *FhMYB27* and *FhMYBx* were overexpressed in *Arabidopsis*, a Rosids clade representative. Both FhMYB27 and FhMYBx could not only repress anthocyanin biosynthesis, but also resulted in glabrous phenotype of *Arabidopsis* (Supplementary Figs. 2 and 3). Moreover, *Arabidopsis* seeds overexpressing *FhMYB27* demonstrated transparent testa, which was derived from the low PA accumulation (Supplementary Fig. 2). Similarly, *FhMYB27* and *FhMYBx* were also overexpressed in Asteroids clade plant *Nicotiana tabacum*. As results, pale red or white flowers were observed in tobaccos overexpressing *FhMYB27*. Meanwhile, reduced anthocyanins, PAs and related genes were accordingly detected (Supplementary Fig. 4). However, no obvious phenotype was observed in tobacco plants overexpressing *FhMYBx*.

**FhMYB27 and FhMYBx interact with FhTT8L.** The alignment of FhMYB27 or FhMYBx and other MYB repressors had demonstrated the conserved bHLH-binding motif in the R3

domains, which implied the potential interactions between FhMYB27 or FhMYBx and bHLH factors. To probe the functional mechanisms of FhMYB27 and FhMYBx, they were subjected to yeast two hybrid and transient protoplast assays to clarify the correlations between the MYB repressors and MBW components. The interactions between any two factors were firstly evaluated by the formerly reported CrY2H-seq system[55]. The results corroborated the previously studied MBW activator complex consisted of FhPAP1, FhTT8L and FhTTG1[48]. Moreover, the yeast clones generated from the mating yeast strains expressing DB-fused FhTT8L and AD-fused FhMYB27 or FhMYBx preliminarily indicated the interactions between the bHLH factor and MYB repressors (Supplementary Fig. 5). The interactions were also assessed by the formerly used Gal4: GUS system[56,57]. The co-transfection of Gal4: GUS and GD-fused *FhTT8L*, *FhTTG1*, *FhMYB27ΔC* or *FhMYBx* showed no significant *GUS* expression (Fig. 3c)[39,49,50]. However, high GUS activity was detected when GD-tagged *FhTT8L* and the reformed transactivator *FhMYB27Δ C-VP16* or *FhMYBx-VP16* were cotransfected together as effectors, indicating FhTT8L might be the bridge linking the MYB repressors to MBW complex (Fig. 4a). Concurrently, the characterized interactions between *Freesia* bHLH regulator and MYB repressors were further confirmed by BiFC assays that bright GFP signal could only be captured between FhTT8L and FhMYB27 or FhMYBx (Fig. 4b).

To validate whether the interactions were conserved when the *Freesia* repressors were heterologously expressed, the convenient Gal4: GUS system was employed to detect the relationship between FhMYB27 or FhMYBx and *Arabidopsis* or tobacco bHLH regulators. As results, FhMYB27 interacted with the anthocyanin related bHLH regulators in either *Arabidopsis* or tobacco, whereas FhMYBx only interacted with *Arabidopsis* bHLH factors (Supplementary Fig. 6), which might elucidate the phenomenon that no obvious phenotype was observed in *FhMYBx* overexpressed tobaccos. Arbitrarily, the negative regulation of the anthocyanin biosynthesis by FhMYB27 and FhMYBx in plants seemed to depend on the MYB-bHLH interactions.

**FhMYB27 and FhMYBx function through MBW complex.** To determine if FhMYB27 or FhMYBx could repress anthocyanin biosynthesis related genes directly, the *Freesia Fh3GT1* promoter-GUS fusion construct was employed in the promoter activation assays. In our previous studies, *Fh3GT1* promoter could be efficiently activated by FhPAP1 solely or in combination with FhTT8L[41,48]. To unveil the negative regulation by FhMYB27 or FhMYBx, the positive regulator FhPAP1 or FhTT8L was also used together with the repressor construct, and the reduction in activation was measured by relative GUS activity. As results showed in Fig. 5a, FhMYB27 alone or cotransformed with FhTT8L would not give rise to obvious GUS activity regardless of whether the C-terminal repression domains existed or not. As expected, FhMYB27 could dramatically repress the transactivation of *Fh3GT1* promoter activated by FhPAP1 and FhTT8L. However, the repression activity of FhMYB27 would be severely affected by the C-terminal deletion, indicating the critical roles of its repression domains. To further identify whether FhMYB27 could bind DNA sequence directly, FhMYB27ΔC-VP16 was also employed. If FhMYB27 targeted the *Fh3GT1* promoter directly, fusing the strong activation domain VP16 to the reformed FhMYB27 that lacked the repression domain would result in significant promoter activation. Nevertheless, FhMYB27ΔC-VP16 was unable to activate the *Fh3GT1* promoter, with or without the FhTT8L partner. Cotransformation of FhMYB27ΔC-VP16 with FhPAP1 and FhTT8L even led to slightly higher GUS activity

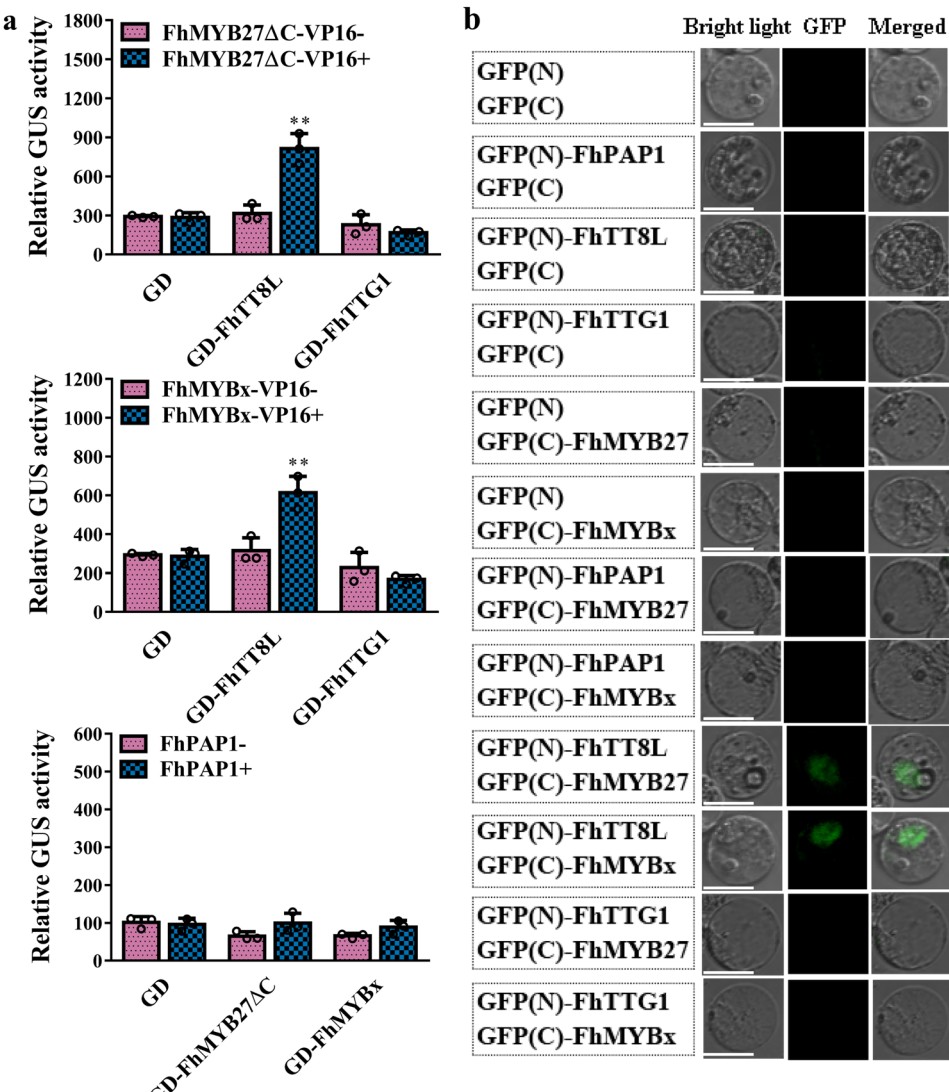

**Fig. 4 FhMYB27 and FhMYBx affected MBW complex forming by interacting with bHLH regulators detected in protoplasts. a** Protein–protein interactions between FhMYB27 or FhMYBx and the MBW components detected by Gal4: GUS system in *Arabidopsis* protoplasts. FhMYB27ΔC indicated the modified version of FhMYB27 whose C-terminus was removed. GUS activities were detected after protoplasts incubation for 21 h. Data represented the mean ± SD of three biological replicates. Student's *t*-test was used to analyze the significant difference (* $p < 0.05$; ** $p < 0.01$). **b** BiFC analysis of the interactions between FhMYB27 or FhMYBx and the MBW components in *Freesia* protoplasts. The fluorescence was observed under fluorescence microscope after incubation for 21 h. Bars indicate 25 μm.

than that of FhPAP1 and FhTT8L. The above results demonstrated that FhMYB27 acted by binding to MBW complex and its repression domains were of great importance. For comparison, the constructs encoding the R3 MYB repressor FhMYBx was also checked. Similarly, FhMYBx could not target promoter sequence and only modestly suppress *Fh3GT1* promoter activated by FhPAP1 and FhTT8L (Fig. 5b). Meanwhile, the functional mechanisms of FhMYB27 and FhMYBx were also validated similarly in another promoter activation assays against *Arabidopsis AtDFR* promoter (Supplementary Fig. 7). Taken the results together, the MBW activator components, especially FhTT8L functioned indispensably in the negative regulation of anthocyanin biosynthesis by FhMYB27 and FhMYBx. The C-terminal repression domains conferred great repression capacity to FhMYB27.

To further understand the roles of C2 and C5 repression domains in FhMYB27, the FhMYB27mC2, FhMYB27mC5 and

FhMY27mC2C5 mutations were generated as described by Yoshida et al., 2015[26]. After *LexA-Gal4: GUS*-based protoplast transfection assay, the mutations could not significantly affect the intrinsic repression capacity of FhMYB27 (Fig. 5c), as well as the negative regulation of FhMYB27 on *Fh3GT1* promoter (Fig. 5d). To validate whether the results were derived from the incomplete mutations of C2 or C5 repression domains, we further generated FhMYB27mC2a, FhMYB27mC5a and FhMYB27mC2aC5a, respectively. Consequently, more site mutations in C2 or C5 domains resulted in relatively higher GUS activities, indicating C2 and C5 domains conferred partial rather than whole transrepression capacity to FhMYB27 in negatively regulating *Fh3GT1* promoter (Fig. 5c, d).

**FhMYB27 and FhMYBx act in anthocyanin regulatory network.** As illustrated, FhPAP1, FhTT8L and FhTTG1 could form the MBW activation complex to activate ABGs expression[48].

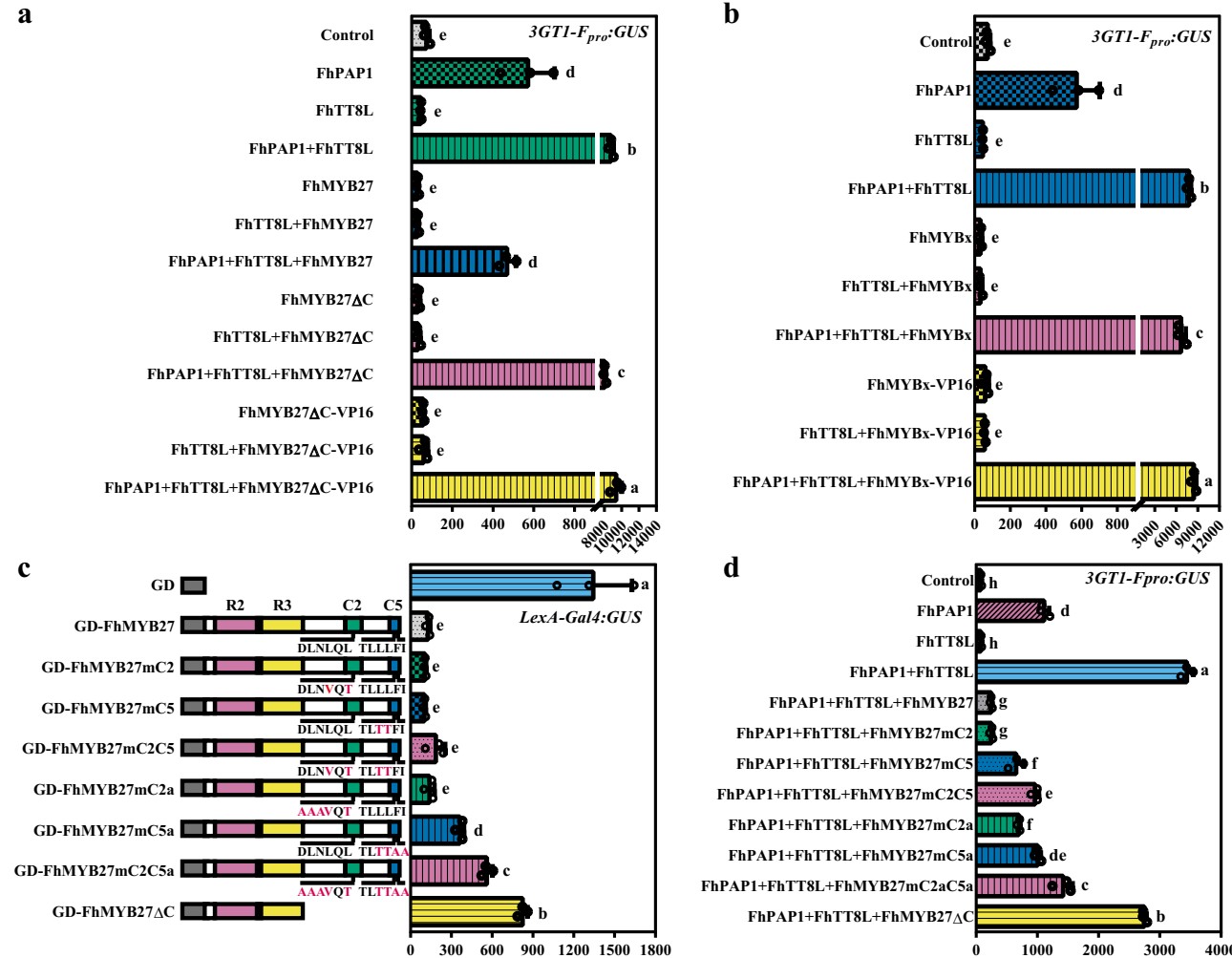

**Fig. 5 FhMYB27 and FhMYBx were corepressors of the MBW activation complex. a**, **b** and **d** Activation and repression assays were performed upon the promoter of the *Freesia* anthocyanin biosynthetic gene *3GT1-F_pro*. The reporter construct contained *GUS* reporter gene driven by *Fh3GT1* promoter. The combinations of effector constructs were diagrammed under the pillars. The reporter and effector constructs were cotransfected into *Arabidopsis* protoplasts. After 21 h incubation, GUS activities were detected. **c** The relative transrepression capacities of mutated FhMYB27 proteins upon *Gal4: GUS* in *Arabidopsis* protoplasts. Modified versions of MYB repressors were assayed: FhMYB27ΔC (C-terminal repression domain removed), FhMYB27mC2 (C2 domain partially mutated), FhMYB27mC5 (C5 domain partially mutated), FhMYB27mC2C5 (C2 and C5 domains partially mutated), FhMYB27mC2a (C2 domain completely mutated), FhMYB27mC5a (C5 domain completely mutated), FhMYB27mC2aC5a (C2 and C5 domains completely mutated). GUS activities were detected after protoplasts incubation for 21 h. Data represented the mean ± SD of three biological replicates. One-way ANOVA was used to find the statistical differences (Duncan's test, $p < 0.05$).

The FhMYB27 and FhMYBx suppressed ABGs expression by interfering the MBW complex aforementioned. To further examine the hierarchical regulation among the MBW factors and MYB repressors, promoter activation assays were conducted. The predicted promoters of *FhPAP1* (1105 bp), *FhTT8L* (755 bp), *FhMYB27* (1172 bp) and *FhMYBx* (1443 bp) were isolated from *Freesia* cultivar Red River® and used in the promoter assays. The promoters of *FhPAP1*, *FhTT8L*, *FhMYB27* and *FhMYBx* were activated by the R2R3 MYB activator FhPAP1 alone or in combination with the bHLH factor FhTT8L (Fig. 6a). However, this activation could be reversed by coexpressing *FhMYB27* or *FhMYBx*. Comparably, FhMYBx depicted moderate suppression on the tested promoters than FhMYB27. To further examine the FhPAP1 binding sites, each promoter was truncated in proportion and subjected to promoter activation assays which revealed the minimized effective promoters of *FhPAP1* (−1105~−785 bp), *FhTT8L* (−438~−280 bp), *FhMYB27* (−330~−170 bp) and *FhMYBx* (−372~−186 bp), respectively (Fig. 6b). Specific

primers were designed according the minimal promoter sequence for ChIP-qPCR analysis. Consequently, multifold enrichments were detected by the ChIP assay *in vivo*, supporting the direct binding of FhPAP1 to the promoters (Fig. 6c). Collectively, a sophisticated regulatory mechanism involving the hierarchical and feedback regulation by activators and repressors should be integrated in *Freesia* anthocyanin biosynthesis.

## Discussion

For more than a century, the biosynthesis and regulation of anthocyanin pigments has been the hotspot and core of specialized metabolites and critically influenced academic dialogues on plant molecular biology, such as the successive blowouts of the first plant transcription factor[58], co-suppression in transgenic plants[59], epigenetic phenomena[60] and transposon discovery[61]. Anthocyanin related analysis continues to provide new insights into new areas, especially the evolution of biochemical pathways and regulatory networks. Here, we characterized two MYB

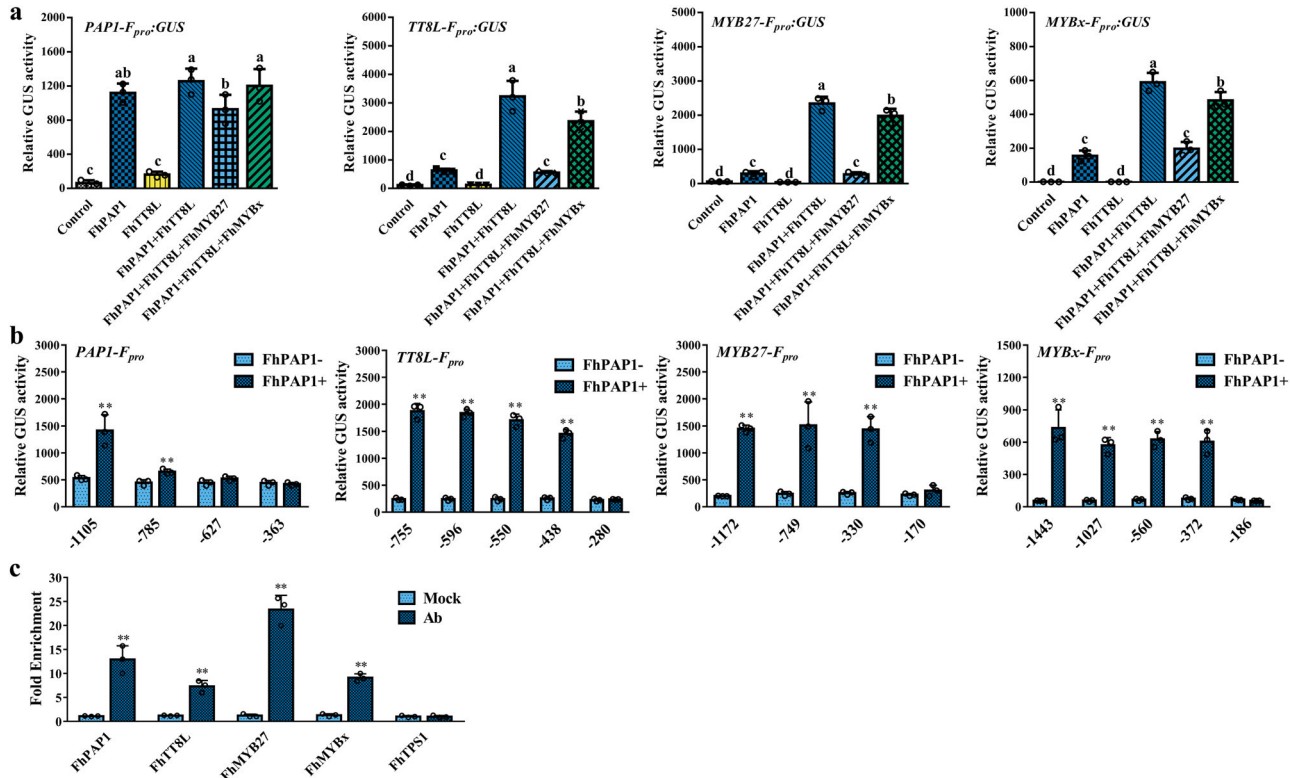

**Fig. 6 The mutual regulations between _Freesia_ anthocyanin regulators. a** The activation assays of MYB-bHLH complex upon the promoter of _Freesia_ _PAP1-F_$_{pro}$, _TT8-F_$_{pro}$, _MYB27-F_$_{pro}$ or _MYBx-F_$_{pro}$ using _Arabidopsis_ protoplasts isolated from _tt8gl3egl3_ triple mutant. **b** The activation assays of FhPAP1 upon differently truncated promoters of _Freesia_ _PAP1-F_$_{pro}$, _TT8-F_$_{pro}$, _MYB27-F_$_{pro}$ or _MYBx-F_$_{pro}$ using _Arabidopsis_ protoplasts isolated from _tt8gl3egl3_ triple mutant. The number indicated the length of each truncated promoter upstream the initiation condon "ATG". The reporter and effector constructs were cotransfected into _Arabidopsis_ protoplasts. The effector constructs expressed each transcription factor from a _CaMV35S_ promoter. **c** The direct binding of FhPAP1 to the promoter of _Freesia_ _PAP1-F_$_{pro}$, _TT8-F_$_{pro}$, _MY27-F_$_{pro}$ or _MYBx-F_$_{pro}$ checked by ChIP-PCR. Data represented the mean ± SD of three biological replicates. One-way ANOVA was used to compare statistical differences in **a** (Duncan's test, $p < 0.05$) and student's _t_-test to analyze the significant differences in **b** and **c** (* $p < 0.05$; **, $p < 0.01$).

repressors FhMYB27 and FhMYBx from monocot _F. hybrida_, and demonstrated that they could down-regulate anthocyanin metabolism based on both transgenic _Freesia_ petals or protoplasts and transient expression or promoter activation assays, revealing the hierarchical and feedback anthocyanin regulation might be relatively conservative in angiosperm plants.

MYB proteins with vital roles in developmental events have been observed in all eukaryotic organisms. In plant anthocyanin biosynthesis, the widely known MYBs function as activators through MBW complex. Also, two distinct groups of MYB proteins, R2R3-MYBs and R3-MYBs, that participate in anthocyanin biosynthesis as repressors have been identified. In _F. hybrida_, FhMYB5 and FhPAP1 have been found to be general flavonoid pathway related and anthocyanin specific R2R3-MYB activators, respectively[48,49]. In current study, two other MYB factors FhMYB27 and FhMYBx were isolated and cloned based on the well-constructed _Freesia_ transcriptomic database. Bioinformatic and phylogenetic analysis inferred that FhMYB27 and FhMYBx were orthologs of petunia MYB27 and MYBx, respectively (Fig. 1). Further analysis revealed that _FhMYB27_ and _FhMYBx_ expression correlated with anthocyanin pattern or ABGs expression, with high transcript abundance in colored tissues or organs (Fig. 2). Such positive correlations in developmental or tissue specific expression were also observed with FaMYB1, PtrMYB57 and PhMYBx orthologs[32,62,63]. On the other hand, a group of MYB repressors showed opposite expression patterns or negative correlations with anthocyanin accumulation or positive

regulation, such as VvMYBC2-L3, PtrMYB182, PhMYB27 and AtMYBL2 repressors[24,26,32,64,65]. The differences in expression patterns of these MYB repressors might reflect their undetermined roles in preventing ectopic accumulation or feedback regulation of anthocyanin biosynthesis[66]. Our further data from _FhMYB27_ and _FhMYBx_ overexpressed petals clearly indicated that anthocyanin was decreased and transcripts of relevant anthocyanin genes were suppressed in transgenic _Freesia_ protoplasts (Fig. 3). Alternatively, the whole-plant transformations with _FhMYB27_ or _FhMYBx_ were conducted in _Arabidopsis_ and tobacco, revealing their repression functions in anthocyanin biosynthesis (Supplementary Figs. 2–4). Collectively, FhMYB27 and FhMYBx, orthologs of _Arabidopsis_ AtMYBL2 and AtCPC, _Petunia_ PhMYB27 and PhMYBx respectively, are MYB repressors of anthocyanin biosynthesis in _F. hybrida_, and their ectopic expression in eudicot plant mimicked the phenotypes of other dicotyledonous MYB repressors, demonstrating their conserved roles in these plants.

Though FhMYB27 and FhMYBx were anthocyanin repressors, FhMYB27 showed higher transrepression capacity than FhMYBx in regulating anthocyanin biosynthesis. This conclusion could be drawn from the following aspects; first, FhMYB27 possessed C2 and C5 repression domains involved in the negative control of flavonoid pathway (Fig. 1a)[24,64,67]. Second, GD-tagged FhMYB27 demonstrated stronger repression activity than FhMYBx when cotransfected with LexA-Gal4: GUS and LD-VP16 in protoplasts (Fig. 3b). Third, FhMYB27 could not acquire transactivation

ability solely by being fused to the activation domain VP16 unless the deletion of C2 and C5 repression domains aforementioned was simultaneously conducted (Fig. 3c). Fourth, FhMYB27 gave rise to more critically reduction of ABGs expression when transformed into *Freesia* protoplasts at same dose, which was in concomitant with the phenotypic changes in transgenic petals, whereas the reduction effect of FhMYBx was largely affected by the ratio of activators to repressors (Fig. 3e). Fifth, *Arabidopsis* overexpressing *FhMYB27* showed other phenotypic abnormalities such as the transparent testa in contrast to those expressing FhMYBx (Supplementary Fig. 2). The discrepancy in repression abilities of different MYB repressors were also observed in other plants, hinting a fine-tuned and integrative network of anthocyanin biosynthesis might comprise regulators with different characteristics[10,26].

To date, all the anthocyanin relevant MYB repressors contain the conserved R3 domain at the N terminus, as well as the bHLH-interacting [D/E]Lx$_2$[R/K]x$_3$L$x_6$Lx$_3$R motif in R3 domain[68]. The intermolecular interactions between MYB repressors and bHLH factors were widely confirmed in numerous angiosperm plants[10,27,65]. Furthermore, when the bHLH-interacting sequences of MdMYB15L and PtrMYB182 were mutated, the interactions with their IIIf bHLH partners were blocked and their repression roles on anthocyanin biosynthesis were lost, indicating the critical roles of bHLH-binding sequence in MYB repressors[26,69]. Likewise, the bHLH-interacting sequences were also found conserved in FhMYB27 and FhMYBx (Fig. 1a), which was supposed to play indispensable roles in the following detected intermolecular interactions (Supplementary Fig. 5; Fig. 4).

In addition to the conserved bHLH-binding sequence in the R3 domain of these MYB repressors, the C2 motif containing the core sequence LxLxL or DLNxxP has also been found in most of the R2R3-MYB repressors, i.e. PhMYB27, PtrMYB57 and MdMYB16, whose repression activities could be reduced or deprived by the deletion or mutation of the C2 motif[10,27,63]. Moreover, another flavonoid repression signature originally characterized from *Arabidopsis* AtMYBL2 named TLLLFR (C5) are also observed in some repressors, such as VvMYBC2 and VvMYB4-like. Interestingly, both C2 and C5-like motifs were found in FhMYB27. The C-terminal truncation or point mutation assays showed decrease of transrepression capacity of FhMYB27 in varying degrees, illustrating the nonnegligible roles of C2 and C5 motifs for FhMYB27 in its anthocyanin regulation (Fig. 5). Collectively, both the bHLH-interacting motif and the repression motif are important for FhMYB27 in suppressing ABG expression. The results were consistent to PhMYB27 but comparatively different with PtrYB182 which showed high sequence similarity with FhMYB27 and also contained the conserved C2 and C5 motifs, however. Functional analysis of PtrMYB182 indicated that it was the bHLH-interacting site rather than the repression motifs exerted more important roles for its repression activity[26], which would also have been speculatively postulated from the incomplete mutations of C2 or C5 domain if FhMYB27mC2a, FhMYB27mC5a or FhMYB27mC2aC5a had not employed.

As results intricately concluded in Fig. 5 and Supplementary Fig. 7, FhMYB27 and FhMYBx functioned consistently like PhMYB27 and PhMYBx in eudicot plants *Petunia hybrida*[10]. The binding to MBW complex bridged by bHLH factor FhTT8L seemed to be indispensable to MYB repressors, whereas the repression domains were also essential for FhMYB27. Previously, repressors have been categorized into FaMYB1-like repressor that is incapable to identify target promoters such as PhMYB27, and AtMYB4-like factors that could bind to target promoter directly such as MdMYB16[10,27]. Obviously, FhMYB27 belonged to FaMYB1-like type either from the bioinformatic analysis or promoter assays, whose target genes largely depended on the blocked MBW complex (Figs. 1 and 5).

Landmark studies have revealed that the anthocyanin relevant MYB activators, bHLH factors, WD40 proteins and MYB repressors are integrated into a hierarchical and feedback network in eudicot plants[10,26,28–30,65]. The direct activations of *FhTT8L*, *FhMYB27* and *FhMYBx* by FhPAP1 indicated the reciprocal regulation among the MBW components in *F. hybrida* (Fig. 6), and a hierarchical and feedback regulatory network of *Freesia* anthocyanin biosynthesis was herein proposed by an adaptation from Albert et al., 2014[10] (Fig. 7). As flower development, FhPAP1 and FhTTG1 were progressively expressed and FhPAP1 could successively activate the ABGs and *FhTT8L* expression. The expressed FhTT8L could then interact with FhPAP1 and FhTTG1 to form MBW complex which could in return positively strengthen the transcripts of *FhTT8L* and ABGs and finally resulted in anthocyanin biosynthesis. Both FhMYB27 and FhMYBx could also be activated by MBW complex to fine-tune floral anthocyanin biosynthesis in *F. hybrida*. FhMYB27 could bind to MBW complex by interacting with FhTT8L, then transform MBW activator complexes into repressor ones. Comparatively, FhMYBx mainly functioned by titrating FhTT8L to decrease the dose of MBW activator complex. As narrated recently, the cooperation of MYB activator NEGAN and repressor RTO resulted in the spotted pigmentation patterns in *Mimulus* flowers[15]. There are also various pigmentation patterns including spotted pigmentation in *Freesia* flowers, and the similar subcellular localization of R3 MYB repressors and feedback regulation intrigue us to further investigate how the interactions of *Freesia* regulators affect the flower pigmentation patterns. Overall, our results implied that the anthocyanin regulation network in *F. hybrida* was much similar to that previously reported in other eudicots, such as petunia, *Arabidopsis*, *Mimulus*, poplar and grapevine[10,15,26,28,30,65], indicating the hierarchical and feedback network might be conserved in angiosperm plants.

Studies of anthocyanin regulation network from *F. hybrida* may be precursor, to our knowledge, for advanced investigations in monocots, especially in flowering plants from *Iridaceae* family. The results elaborate the regulatory mechanism underlying flower pigmentation in *F. hybrida* and contribute to our understanding of anthocaynin regulation among plants at different evolutionary positions.

## Methods

**Plant materials and growth conditions**. *F. hybrida* cultivar Red River® and Ambiance were grown in a greenhouse with a photoperiod of 14 h light and 10 h dark at 15 °C. For gene expression analysis, flowers of Red River® from different individual plants were sampled at different developmental stages and the fully blooming flowers were divided into toruses, calyxes, petals, stamens and pistils. Moreover, roots, leaves and scapes were also collected as earlier defined[49]. All the samples were drenched into liquid nitrogen immediately and kept at −80 °C until used. The young inflorescence segments of Red River® were used to induce calluses in callus proliferation medium following the method described earlier[47]. The *Freesia* protoplasts were isolated from the pale-yellow nodular calluses using well-established methods[47,49,50].

*A. thaliana* (*Columbia-0*) and *N. tabacum* (*cv. K326*) were cultivated in the greenhouse at 22 °C for 16 h light and 8 h dark. Wild-type and transgenic leaves of four-week-old *Arabidopsis* were used to detect anthocyanin and PA accumulations and gene expression levels. About 4-week-old *Arabidopsis* leaves were harvested for protoplasts isolation. The fully blooming wild-type and transgenic tobacco flowers were further divided into petals, stamens, pistils, toruses and calyxes as earlier defined[49,50].

**DNA, RNA, cDNA preparation and gene expression analysis**. NuClean Plant Genomic DNA Kit and OminiPlant RNA Kit (both from CWBIO, Beijing, PRC) were employed for genomic DNA and total RNA extraction, respectively. 500 ng of total RNA were further reversely transcribed into cDNA using UEIris II RT-PCR System for First-Strand cDNA Synthesis Kit (US Everbright® Inc., Suzhou, PRC)

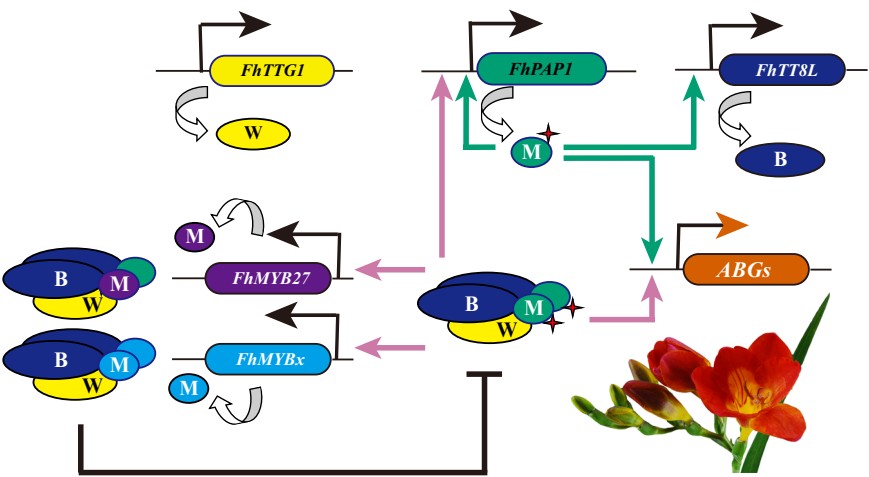

**Fig. 7 Proposed transcriptional regulation model of the anthocyanin biosynthetic genes in *Freesia* cultivar Red River®.** *FhPAP1*, *FhTT8L* and *FhTTG1* were expressed as *Freesia* flower development. In Red River®, FhPAP1 could activate the expression of ABGs, *FhPAP1* and *FhTT8L*. The expressed FhTT8L could then interact with FhPAP1 and FhTTG1 to form MBW complex. Subsequently, the MBW complex could in return positively strengthen the transcripts of *FhTT8L*, as well as highly activate the ABGs, which resulted in anthocyanin biosynthesis. Additionally, two kinds of repressors, FhMYB27 and FhMYBx, could also be activated by MBW complex with the purpose of balancing anthocyanin biosynthesis in *Freesia* flowers. Solid arrows with bluish green and reddish purple represented transcriptional regulation. W, WD40 protein FhTTG1; M, MYB regulator FhPAP1; B, bHLH factor FhTT8L, ABGs, anthocyanin biosynthetic genes.

following the manufacture's instruction. Transcript levels were detected by qRT-PCR using SYBR Green Master Mixture (US Everbright® Inc., Suzhou, PRC) with parameters narrated elsewhere[70].

**Gene and promoter clone.** The petunia PhMYB27 and PhMYBx were used as bait sequences to BLAST the potential anthocyanin biosynthesis related MYB repressors against the constructed *Freesia* transcriptomic database using TBLASTN algorithm[70]. The obtained sequences were further analyzed by manual BLASTX search of National Center for Biotechnology Information (NCBI, https://www.ncbi.nlm.nih.gov/). The specific primers were designed to clone the candidate genes accordingly (Supplementary Dataset 1). The amplicons were subsequently ligated into *pESI-T* vector by Hieff Clone® Zero TOPO-TA Cloning Kit (Yeasen, Shanghai, China) for further sequence confirmation.

For protein alignment and phylogenetic analysis, flavonoid biosynthesis related MYB regulators were retrieved from Genbank (http://www.ncbi.nlm.nih.gov/Genbank/). The sequence alignment was subjected to Clustal Omega algorithm (http://www.ebi.ac.uk/Tools/msa/clustalo/), then transferred to MEGA Version X to generate the phylogenetic diagram[71].

The promoters of structural genes participating in anthocyanin biosynthesis had been cloned and used in earlier studies[48,49]. Genome Walking Kit (Takara, Dalian, PRC) was used for promoter isolations of *FhPAP1*, *FhMYB27* and *FhMYBx* as outlined by the manufacturer and the potential sequences were cloned into *pESI-T* vector for sequence confirmation.

**Constructs.** To perform transient protoplast assay, human influenza hemagglutinin (HA)- or GD-tagged *FhMYB27* and *FhMYBx* vectors were constructed by Minerva Super Fusion Cloning Kit (US Everbright® Inc., Suzhou, PRC) as mentioned in earlier work[70]. The HA and GD-tagged *FhMYB27* were site-mutated at C2 or C5 repression domain singly or together following the Fast Mutagenesis System (TransGen Biotech, Beijing, China). The HA or GD-tagged *FhMYB27ΔC* was cloned by deleting the C terminus of *FhMYB27* using specific seamless cloning primers (Supplementary Dataset 1). Moreover, the strong activation domain VP16 (viral activation domain) fused constructs were generated by substituting the respective termination codon with VP16[54]. For constructs used in bimolecular fluorescence complementary (BiFC) and subcellular localization assays, the respective open reading frame of *FhMYB27*, *FhMYBx* or *FhPAP1* was subcloned to substitute *FhTT8L* in the earlier used vector *35 S: GFPN/GFPC/GFP-FhTT8L*. The *35 S: FhMYB27/FhMYBx-GFP* vectors were also constructed by fusing the GFP to the C-terminus of each protein. For constructing the vectors used in plant transformation, HA-tagged *FhMYB27* or *FhMYBx* was digested by *EcoR* I and cloned into the binary vector *pPZP211*. Otherwise, *pADlox* and *pDBlox* used in the earlier reported Cre (Cyclization recombination enzyme)-reporter-mediated yeast two-hybrid (CrY2H) were modified by adding the restriction sites of *Nde* I and *Sac* I in attR1 and attR2 site, respectively[55]. The *Freesia* regulators were seamlessly cloned into the modified *pADlox* and *pDBlox* at the restriction sites of *Nde* I and *Sac* I to produce prey and bait vectors employed in yeast two-hybrid assay. Constructed vectors were ascertained by sequencing and primers used are provided in

supplementary material (Supplementary Dataset 1). Other constructs used in present studies were described in our earlier studies[48–50].

**Plant transformation.** For transient gene expression in *Freesia* cultivar Ambiance, *Agrobacterium* strain GV3101 with different vectors were injected into the 20-30 mm long buds following Sparkes et al., 2006[72]. The petals were harvested and frozen in liquid nitrogen in three days for further analysis. About 6-week-old *Arabidopsis* was transformed by GV3101 containing the binary vectors over-expressing *FhMYB27* or *FhMYBx* following the floral dip method[73]. The transgenic seeds were selected by 50 ng μL⁻¹ kanamycin on 1/2 MS medium and T3 seedlings were further analyzed. The stably transformed tobacco was generated using protocol explained by Sparkes et al., 2006[72].

**Anthocyanin and proanthocyanidin analysis.** Anthocyanins and PAs were quantified in both wild-type and transgenic samples according to the earlier described methods[49]. For comparison of the PA contents in wild-type and transgenic *Arabidopsis* seeds, T3 seeds were stained by 0.3% (w/v) DMACA (dimethylaminocinnamaldehyde) following the methods mentioned in earlier studies[50]. The phenotypes were captured by a dissecting microscope.

**Transient protoplast transfection assay.** The detailed procedures for plasmid preparation and transient transfection assays were described elsewhere[47,54]. Briefly, all the constructs used in transient transfection assays were extracted by endotoxin-free GoldHi EndoFree Plasmid Maxi Kit (CWBIO, Beijing, PRC). The plasmids used in PEG-mediated transfection were concentrated to an estimated final concentration of 3–4 μg μL⁻¹ using isopropanol and NaCl. The protoplasts were isolated from 4-week-old sub-cultured *Freesia* calluses or 3 to 4 weeks old *Arabidopsis* rosette leaves by enzyme solution containing Cellulase R-10 and Macerozyme R-10 (both from Yakult Pharmaceutical Ind. Co., Ltd., Japan). Combinations of 10 μg aliquot of plasmids were transfected into protoplasts by PEG3350 (Solarbio Life Science, Beijing, PRC). After incubation in the darkness for 20–21 hours at room temperature, the protoplasts were centrifuged and subjected to further analysis.

**GUS activity measurement.** The measurement of GUS activities was performed following Yoo et al., 2007[54]. Briefly, the protoplasts were collected by centrifugation at 120 × g for 3 min. The Luciferase Cell Culture Lysis 5 × Reagent (Promega, Madison, WI) was added to rupture the protoplasts. 10 μL of the protoplast lysate was reacted with 100 μL of MUG (4-methylumbelliferyl glucuronide; Gold Bio-Technology, Inc.) substrate mix containing 1 mM MUG and 2 mM MgCl₂ in 10 mM Tris-HCl (pH 8.0) for 1 h. The GUS activities were detected by Synergyrgyaimicroplate reader (Bio TEK) after 100 μL of 0.6 M Na₂CO₃ was added to stop the reaction.

**Chromatin immunoprecipitation (ChIP) assay.** ChIP assays were performed using EpiQuik™ Plant Chromatin Immunoprecipitation Kit (Epigentek, Farmingdale, USA) according to the instruction. Briefly, the *Agrobacterium* strains

GV3101 containing *HA-FhPAP1* were cultured and suspended in the infiltration medium[72]. The *Agrobacterium* was subsequently injected into the Ambiance buds with 20–30 mm long. The buds that bloomed in 48 hours with injected petals were harvested and linked by dipping in 1% formaldehyde. The samples were sonicated after being powdered in liquid nitrogen to make soluble chromatin solution, then combined with the anti-HA antibodies pre-embedded in the microwells. The microwells immobilized with normal mouse IgG were used as the negative control. The precipitated DNA was then reversed by purification process as instructed by the manufacturer. The purified DNA was used for further qRT-PCR analysis.

**qRT-PCR analysis**. Specific primers were designed for detecting *FhMYB27* and *FhMYBx* transcripts and ChIP-PCR analysis (Supplementary Dataset 1), while other primers for qRT-PCR analysis were found in earlier studies[48]. The relative gene transcripts were normalized by *Freesia 18 S rRNA*, *Arabidopsis Actin* and *Nicotiana tubulin* before being calculated by the $2^{-\triangle\triangle CT}$ formula[74]. The relative expression data of ABGs from *Freesia* were changed to $\log_2$ values and processed by HemI 1.0 to produce the heatmap[75]. The ChIP signals relative to no-antibody control were processed and designated as the fold enrichment.

**Yeast two-hybrid assay**. The formerly reported CrY2H-seq system was employed after minor modification in testing the potential MYB-bHLH-WD40 interactions[55]. Succinctly, the bait vectors having Gal4 binding domain and prey vectors containing Gal4 activation domain were transformed into CRY8930 and Y8800 yeast strains, respectively. The transformants selected on synthetic complete dropout (SD) media were inoculated into the corresponding selective liquid medium overnight at 30 °C. A 20 μL aliquot of donor and host strains were transferred to100 μL of YPDA media in a fresh 2-mL 96-well plate and incubated at 30 °C by shaking overnight. The mating products were 10-fold diluted by water and a 3 μL aliquot was dipped to different selective plates for 3 days at 30 °C.

**Statistics and reproducibility**. Student's *t*-test was computed to analyze the significant difference between two groups. Asterisk and double asterisks indicated values differ significantly with the highest value at < 0.05 and $p < 0.01$, respectively. One-way ANOVA was carried out to compare statistical differences among groups with Duncan's test. Significant differences were marked with the highest value at < 0.05. The materials collected from *F. hybrida* with highly uniform state of growth were sampled from different individuals to form one biological replicate, and three replicates were subjected to gene expression analysis. In detail, more than 5 flowers at each developmental stage from at least 3 individual plants were pooled to form one replicate. Floral tissues detached from Stg 5 flowers were also from more than 3 plants, and each replicate was pooled with more than 10 floral tissues. Total RNA from each replicate was extracted independently for further analysis. For the overexpression assays, the phenotypes were assessed as stable and reproducible when independently appeared in at least 6 lines. The transgenic lines with most obvious or representative phenotypic changes were subjected for further analysis. As for the transient protoplast assay, enough quantities of protoplasts were prepared and divided into three replicates, each replicate of protoplasts was transfected with the same dose of plasmids and deemed as one biological replicate. In order to confirm the accuracy of the transient protoplast assay, protoplasts were isolated and prepared more than 3 times and the transfection were also repeated as mentioned above. Data in present manuscript were derived from the same batch with three biological repeats. The overall tendencies in each assay had been checked among batches of transfection and same tendencies were observed to prove the feasibility of the transfection system. Additionally, all of the qRT-PCR experiments in the study were performed three biological replications using the aforementioned samples.

**Reporting summary**. Further information on research design is available in the Nature Research Reporting Summary linked to this article.

## Data availability

The sequence of *FhMYB27* and *FhMYBx* have been deposited in NCBI under GenBank accession numbers MT210094 and MT210095, respectively. Other data is available in Supplementary Dataset 2.

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

## Acknowledgements

This work was supported by the National Natural Science Foundation of China (31900252, 31972445), the China Postdoctoral Science Foundation funded project (2018M641761), the Department of Science and Technology of Jilin Province (20190201299JC, 20190303095SF), the Programme for Introducing Talents to Universities (B07017) and the Fundamental Research Fund for the Central Universities. The funders had no role in study design, data collection and analysis, decision to publish, or preparation of the manuscript.

## Author contributions

Y.L., X.S., R.G., T.H., J.Z., Y.W. and S.K. performed the experiments and helped analyze data. Y.L. wrote and revised the manuscript with X.G. X.G. designed the experiments and discussed with L.W. All authors have participated in this research and approved the final manuscript.

## Competing interests

The authors declare no competing interests.
