## [Peer Review File · Communications Biology]

Reviewers' comments:

Reviewer #1 (Remarks to the Author):

The manuscript by Li et al examines whether the MBW regulatory network models described for eudicots (Albert et al 2014, Plant Cell) also exists in monocots. This is an interesting question, which until now has not been addressed because of the lack of a suitable model system (as raised in Albert et al 2014 Plant Signaling and Behavior). The grass models, such as rice and maize, are phylogenetically very distinct from other groups of monocots and lack the complex pigmentation patterning that is important for signalling to insect and animal pollinators. This study is both timely and interesting. However, there are areas that need attention.

Major

1) It was conspicuous that there was a lack of citations or discussion of the MBW components that have been identified in non-grass monocots. There are many publications from various orchids, onion, and lilies. This also includes repressors. This need revisiting and appropriate revisions made.

2) The nature of the replication throughout this manuscript is not described. What did each replicate consist of?

3) Figure legends need to contain sufficient detail and information that they can be understood independently from the main text. This was not always the case. For example, Figure 3E is incomprehensible. It isn't even clear how this experiment was performed. (see 8). Please revise.

4) I think the name MYB4 draws an unhelpful link between AtMYB4 from Arabidopsis and FhMYB4. L255 "... to genes encoding orthologues of PhMYB27 and PhMYBx were mined and displayed maximum similarities to Arabidopsis AtMYB4 and AtCPC..."

AtMYB4 and FhMYB4 are not orthologues - the phylogenetic tree and sequence alignments presented in Figure 1 demonstrate this clearly. AtMYB4 belongs to a clade within SG4 that regulate phenylpropanoid genes (e.g. C4H). The putative orthologue from Arabidopsis is AtMYBL2 - this was not included on the tree or analyses. This sequence often fails to correctly clade, due to the loss of the R2 domain, which is within the DNA binding domain, and provides most of the sequence information for building phylogenies. AtMYBL2 also has the TLLLFR repression domain, in addition to the EAR motif - FhMYB4 has these also, as many of the anthocyanin or proanthocyanidin repressors do. I would consider renaming MYB4, to MYBL2 or MYB27. But at minimum, L255 needs to be rewritten to explain what is meant by the similarity to AtMYB4.

5) It would be very helpful to have a picture of the flower of 'Red river' cultivar in figure 2, with the different parts sampled in part b indicated. The names of the flower organs in freesia are not commonly known, so this makes it very difficult for readers to actually know which organ it is. There is also no information about the pigmentation of that tissue - so how can readers know if this is associated with anthocyanin accumulation? Similarly, pictures of the developmental series and the pigmentation status is necessary.

6) Figure 2 A and B also lack key information, such as the expression pattern of the MYB activator, bHLH, WDR and at least a few representative anthocyanin biosynthesis genes. This really is crucial information, and without it, the existence of the gene regulation network is not substantiated.

7) Figure 3d - protoplast infiltration qPCR. There is only the result for when the MYB and bHLH are together, but what about the negative control? Such as transfection only control (GUS, GFP etc).

8) Figure 3e It is not clear what this experiment shows or how it was done. I'm assuming "Ambience" is a white flowered cultivar – but what do we know of this? What is the mutation? The experiment isn't controlled correctly, because it should have a GFP/GUS or similar negative control to show that infiltration with *Agrobacterium* alone is not inducing colour. Then the next experiment is to coexpress the MYB +bHLH (which needs explanation in the results) and then assess whether the repressors inhibit this. What is the "control"? I suspect MYB+bHLH, but this information isn't provided.

9) There are frequent references to unpublished data – this hasn't been acceptable in any journal since the advent of supplementary data in electronic formats. This is because the statements made cannot be validated or critiqued by reviewers or readers. I suspect there is an additional manuscript on the MYB activator gene that is yet to be published. I appreciate the authors are likely developing a different story here, but this current manuscript really requires the other article to be published (or at least accepted and in press) first. If this article is accepted for publication, then I think the in press article would need to be shared with the reviewers and editor to ensure the referencing to that study was appropriate.

10) qRT-PCR was normalised to 18S rRNA. This is quite poor, because there is sufficient sequence information to design a degenerate primer and isolate a fragment of ACTIN, GAPDH or similar. This doesn't guarantee the reference genes would be stable, but it would be a lot better than using a reference gene that is known to be variable and expressed at many orders of magnitudes different to the genes of interest. How was the cDNA synthesis primed? I tried finding the information from the kit used, but was unable to. Was oligo dT used? Random hexamers? A mix? Or was a gene-specific primer for 18S included also? If oligo dT, then the normalisation to 18S rRNA is even more inappropriate, because it would be a contaminant of the RT (rRNAs are not polyadenylated).

11) The model presented in Figure 7 is an adaptation from Albert et al 2014 (*Plant Cell*) and this should be stated. I am not sure including a large anthocyanin biosynthesis pathway assists. It occupies a large amount of space in this figure, but is not the focus of this study. I would recommend removing this, and just focusing on the gene regulation network.

Minor

L61 The article cited does not show that flavonols were found in liverworts. Flavones have been identified in liverworts. Flavones and flavonols appear to perform similar roles, but they are distinct branches from the flavonoid pathway.

L68 I don't think there is still a debate about the function of the WDR/WD40 protein. Since understanding the MYB, bHLH and WDR proteins all interacted together in a complex, it fits that this protein is providing a scaffolding role for the assembly of MBW (+ potentially others) proteins.

Line 69-71. The distinct of bHLH A and B is a new labelling system I have not seen before. I wonder why the authors haven't adopted bHLH1 (R/JAF13/Delila/EGL3) and bHLH2(IN/An1/Mut/TT8), as is used in many articles, including the model from Albert et al 2014 (*Plant Cell*), which this study is largely following.

Line 73 Until (not Till)

Line 84. MYBL2 belongs to the same clade of MYB repressor as PhMYB27, FaMYB1 etc. It is really unhelpful to refer to it as an R3MYB without explanation or qualification. It has had a truncation of the

first exon, which is why the R2 domain is largely missing. But it is phylogenetically and functionally related to the R2R3-MYB repressors, not the typical R3-MYBs. This matters because in the results, FaMYB4 is said to be closer in sequence to AtMYB4 (it is not). It is much closer to AtMYBL2.

Line 86 is missing Albert et al 2011 (Plant Journal) and 2014 (Plant Cell) are required for describing MYBx, MYB27 and the models being described.

Reviewer #2 (Remarks to the Author):

Two conclusions were made in this manuscript: the first one, FhMYB4 and FhCPC, belonged to R2R3-MYB and R3-MYB subgroup respectively, were isolated from Freesia flowers and functionally characterized. Functional studies indicated that they could suppress anthocyanin biosynthesis by inhibiting ABGs expression and might function in differential mechanisms according to repressor domains in the C-terminus; the second one is that two kinds of repressors, FhMYB4 and FhCPC, could also be activated by MBW complex with the purpose of balancing anthocyanin biosynthesis in Freesia. It is useful information to reveal the hierarchical and feedback anthocyanin regulation might be relatively conservative in angiosperm plants. However, there are still some recommends in the manuscript that need to be improved.

- 1) For the methods section, lack of details of some experiments were found in the section, e.g. GUS activities measurement.
- 2) Detailed methods of statistics analysis used in the study were lost.
- 3) Some bootstrap values are relatively low, especially FhMYB4 and FhCPC highlighted with the red stars, which may lead to unconvincing clustering in fig. 1b. The NJ method for construction of the phylogenetic tree is not accurate and unacceptable, usually the ML tree (Maximum likelihood tree) may be a better choice. The gene trees built by different methods such as NJ, ML and MP methods, have similar topological structure, so it can be considered reliable. Furthermore, an outgroup (from mosses or other lower plant species) should be added.
- 4) There is a serious problem that the results of GD, GD-FhMYB4 and GD-FhCPC is inconsistency between Fig. 3b and Fig. 3c.
- 5) As far as I know, unpublished data cannot be used as a reference, but the author cites these documents more often.
- 6) All species names require italics, such as *Arabidopsis thaliana*, *Marchantia*, *Freesia*, etc.
- 7) The general rule, when the Latin name of species appears for the second time in the text, the generic name is usually abbreviated, for example, *A. thaliana*. There are many such writing errors in the text.
- 8) Through a series of experiments, the author tries to prove that both novel MYB repressors that involved in the inhibition of anthocyanin synthesis in monocot is evolutionarily conservative as in dicot plant. However, more comparison between orthologous from dicot plant, for example, from *Arabidopsis* should be supplied.
- 9) More editing is needed to improve the organization of the text and English style, and to avoid inappropriate citations.

Reviewer #3 (Remarks to the Author):

Flower color pattern that resulted from anthocyanin is not only visually appealing, but also have implications in health and agriculture. Li, Shan and their colleagues reported two MYB anthocyanin

repressors in monocot *Freesia hybrida*. One is an R2R3 MYB type, another is an R3 MYB type. Through a series of painstaking experiments *in vivo* and *in vitro*, they convincingly demonstrated their two MYBs are true anthocyanin repressors. They found that these MYBs involved in anthocyanin regulation by interacting with the MBW complex. This MBW complex is also involved in a feedback regulatory loop, such as that reported in dicots. This work is a significant achievement for the authors, particularly these experiments are done in a non model system.

However, there are a few things that I hope the authors can carry on to further improve this manuscript.

1 I disagree with the authors that the anthocyanin regulation is not well studied in monocot. Anthocyanin regulation is very well studied in several monocot crops. Therefore, I think it is an inaccurate description of the current understanding of the anthocyanin research.

2 Though the authors describe these two regulators and functionally validate them in *Arabidopsis* and tobacco, these transcription factors have already been well studied, including the mechanisms that the authors worked out. In other words, even the authors have performed a lot of experiments, not much is really new if you take a broader look of the field. Many of these studies are largely confirmatory.

3 The writing is poorly organized, many writings are not conventional. Please have some native speakers to check the writing. Throughout the manuscript, the authors start describing experiment before telling the purpose of the experiment, which makes it very hard to follow.

For example, the very first thing they did in the manuscript is to isolate the MYB orthologues. It is unclear to me why the authors want to study the MYB repressors in *Freesia hybrida*. Why understanding whether the anthocyanin regulation is conserved between monocot and dicot requires studying MYB repressors, there are lots of anthocyanin repressors other than MYBs. I also noticed the authors cited 27~37. It seems to me that the anthocyanin regulation in *Freesia hybrida* is similar to that in dicots, which would contradict author's claim in the introduction. All experiments are very well executed, but are not articulated to make it easier to understand. I also added a few comments in the attached manuscript and hope it helps.

Reviewers' comments:

Reviewer #1 (Remarks to the Author):

The manuscript by Li *et al* examines whether the MBW regulatory network models described for eudicots (Albert *et al* 2014, Plant Cell) also exists in monocots. This is an interesting question, which until now has not been addressed because of the lack of a suitable model system (as raised in Albert *et al* 2014 Plant Signaling and Behavior). The grass models, such as rice and maize, are phylogenetically very distinct from other groups of monocots and lack the complex pigmentation patterning that is important for signalling to insect and animal pollinators. This study is both timely and interesting. However, there are areas that need attention.

Response: Thank you for your positive and inspiring comments on our manuscript. On behalf of the co-authors, I would like to express our great appreciation to you.

Major

1) It was conspicuous that there was a lack of citations or discussion of the MBW components that have been identified in non-grass monocots. There are many publications from various orchids, onion, and lilies. This also includes repressors. This need revisiting and appropriate revisions made.

Response: Thank you for your professional comment. As you inferred, there are many MBW components characterized from non-grass monocots, such as orchids, onion, and lilies¹⁻⁴. We have modified our manuscript and added the information in introduction. Please find the revisions in current version.

2) The nature of the replication throughout this manuscript is not described. What did each replicate consist of?

Response: Thank you for your constructive suggestion. Following your advice and the journal's request, we have added the "Statistics and reproducibility" section in the manuscript. Please find the current version.

3) Figure legends need to contain sufficient detail and information that they can be understood independently from the main text. This was not always the case. For example, Figure 3E is incomprehensible. It isn't even clear how this experiment was performed. (see 8). Please revise.

Response: Thank you for your constructive comment. We are sorry for our unclear description. Following your suggestion, we have added sufficient details and information in figure legends, ensuring the figures could be understood.

4) I think the name MYB4 draws an unhelpful link between AtMYB4 from *Arabidopsis* and FhMYB4. L255 "... to genes encoding orthologues of PhMYB27 and PhMYBx were mined and displayed maximum similarities to *Arabidopsis* AtMYB4 and AtCPC..."

AtMYB4 and FhMYB4 are not orthologues - the phylogenetic tree and sequence alignments presented in Figure 1 demonstrate this clearly. AtMYB4 belongs to a clade within SG4 that regulate phenylpropanoid genes (e.g. C4H). The putative orthologue from *Arabidopsis* is AtMYBL2 – this was not included on the tree or analyses. This sequence often fails to correctly clade, due to the loss of the R2 domain, which is within the DNA binding domain, and provides most of the sequence information for building phylogenies. AtMYBL2 also has the TLLLFR repression domain, in addition to the EAR motif - FhMYB4 has these also, as many of the anthocyanin or proanthocyanidin repressors do. I would consider renaming MYB4, to MYBL2 or MYB27. But at minimum, L255 needs to be rewritten to explain what is meant by the similarity to AtMYB4.

Response: Thank you for your professional comment. As you suggested, we renamed FhMYB4 and FhCPC as FhMYB27 and FhMYBx, respectively, and L255 was rewritten in the current version.

5) It would be very helpful to have a picture of the flower of 'Red river' cultivar in figure 2, with the different parts sampled in part b indicated. The names of the flower organs in *freesia* are not commonly known, so this makes it very difficult for readers to actually know which organ it is. There is also no information about the pigmentation of that tissue – so how can readers know if this is associated with anthocyanin accumulation? Similarly, pictures of the developmental series and the pigmentation status is necessary.

Response: Thank you for your constructive comment. We have added the *Freesia* pictures containing different developmental stages and tissues or organs. Moreover, the pictures and phenotypes can also be found in our earlier studies⁵⁻⁷.

6) Figure 2 A and B also lack key information, such as the expression pattern of the MYB activator, bHLH, WDR and at least a few representative anthocyanin biosynthesis genes. This really is crucial information, and without it, the existence of the gene regulation network is not substantiated.

Response: Thank you for your careful work. The expression patterns of MYB repressors, MBW components and anthocyanin biosynthetic genes (ABGs) in different developmental stages were included in Fig. 2b in current version. Moreover, we also added the expression analysis of MBW components in different tissues or organs in current manuscript.

7) Figure 3d – protoplast infiltration qPCR. There is only the result for when the MYB and bHLH are together, but what about the negative control? Such as transfection only control (GUS, GFP *etc*).

Response: Thank you for your careful work. *Freesia* protoplasts were isolated from calluses induced from young inflorescence, and the ABGs were supposed to be silenced or had low expression levels. In

order to validate the roles of MYB repressors on these genes, we should elevate their transcripts at first. In our latest paper, FhPAP1 has already been characterized to be a robust factor to activate ABGs expression in *Freesia* protoplast and FhTT8L could highly strengthen the promotion⁶. Herein, we just cotransfected FhPAP1 and FhTT8L to activate the ABGs and deemed FhPAP1 and FhTT8L together as background or control. The technique was widely used and the results have been repeated at least three times. Moreover, we would be pleased to repeat the results following your advice if we got the *Freesia* callus again next year as the calluses were all dead because of the closure of the university during COVID-19. We will collect more inflorescence to induce calluses during next February to March which is the flowering phase of *Freesia*. We hope to get your understanding of our present difficulties and circumstances. Again, thank you for your professional comment.

8) Figure 3e It is not clear what this experiment shows or how it was done. I'm assuming "Ambiance" is a white flowered cultivar – but what do we know of this? What is the mutation? The experiment isn't controlled correctly, because it should have a GFP/GUS or similar negative control to show that infiltration with *Agrobacterium* alone is not inducing colour. Then the next experiment is to coexpress the MYB +bHLH (which needs explanation in the results) and then assess whether the repressors inhibit this. What is the "control"? I suspect MYB+bHLH, but this information isn't provided.

Response: Thank you for your professional comment. We have detailed the figure legends in current manuscript. Ambiance is another cultivar of *F. hybrida* with white flowers (<http://www.sierraflowerfinder.com/en/d/ambiance/7031> or <https://www.vanengelen.com/freesia-ambiance-single.html>), which was also included in our earlier studies^{6,8}. As narrated in our earlier paper, all the anthocyanin biosynthetic genes, as well as FhPAP1 and FhTT8L, have significantly decreased expression levels in Ambiance petals, and the colorless petals of Ambiance might mainly derive from the deficiency of FhPAP1⁶. We strongly agree with you that a GFP/GUS or similar negative control to show that infiltration with *Agrobacterium* alone is not inducing color. The results have been elucidated in detail in our latest manuscript⁶. FhPAP1 was able to induce color to Ambiance petals and FhTT8L could highly strengthen the promotion⁶. Herein, we just cotransfected FhPAP1 and FhTT8L together and deemed them as background or control. Again, thank you for your kind comment.

9) There are frequent references to unpublished data – this hasn't been acceptable in any journal since the advent of supplementary data in electronic formats. This is because the statements made cannot be validated or critiqued by reviewers or readers. I suspect there is an additional manuscript on the MYB activator gene that is yet to be published. I appreciate the authors are likely developing a different story here, but this current manuscript really requires the other article to be published (or at least accepted and in press) first. If this article is accepted for publication, then I think the in press article would need

to be shared with the reviewers and editor to ensure the referencing to that study was appropriate.

Response: Thank you for your kind reminding and positive comment. We are sorry for our frequent references to unpublished data. Fortunately, the manuscript about FhPAP1 has just been accepted and we have revised our manuscript in current version.

10) qRT-PCR was normalised to 18S rRNA. This is quite poor, because there is sufficient sequence information to design a degenerate primer and isolate a fragment of ACTIN, GAPDH or similar. This doesn't guarantee the reference genes would be stable, but it would be a lot better than using a reference gene that is known to be variable and expressed at many orders of magnitudes different to the genes of interest. How was the cDNA synthesis primed? I tried finding the information from the kit used, but was unable to. Was oligo dT used? Random hexamers? A mix? Or was a gene-specific primer for 18S included also? If oligo dT, then the normalisation to 18S rRNA is even more inappropriate, because it would be a contaminant of the RT (rRNAs are not polyadenylated).

Response: Thank you for your professional comment. We agree with you that 18S rRNA would not be appropriate as normalization if only oligo dT were employed in cDNA synthesis. However, the primers we used in cDNA synthesis in present study were a mix of oligo dT and random hexamers. So we thought this might be applicable as also narrated in our earlier studies⁶⁻¹⁰. Moreover, we also isolated potential Actin and Ubiquitin sequences from *Freesia* transcriptomic database and designed specific primers to check the results. Similar expression patterns were observed when different reference genes were employed.

Expression analysis of FhPAP1, FhTT8L, FhGL3L and FhTTG1 in different expression stages. The transcripts were normalized to *Freesia Actin* (a) and *Ubiquitin* (b) genes, respectively.

11) The model presented in Figure 7 is an adaptation from Albert *et al* 2014 (Plant Cell) and this should be stated. I am not sure including a large anthocyanin biosynthesis pathway assists. It occupies a large

amount of space in this figure, but is not the focus of this study. I would recommend removing this, and just focusing on the gene regulation network.

Response: Thank you for your valuable suggestion. We have stated the model was an adaption from Albert *et al.*, 2014¹¹ in the manuscript, and the schematic diagram was modified following your advice.

Minor

L61 The article cited does not show that flavonols were found in liverworts. Flavones have been identified in liverworts. Flavones and flavonols appear to perform similar roles, but they are distinct branches from the flavonoid pathway.

Response: Thank you for your reminding. We are sorry for our carelessness. We have modified the sentence here.

L68 I don't think there is still a debate about the function of the WDR/WD40 protein. Since understanding the MYB, bHLH and WDR proteins all interacted together in a complex, it fit that this protein is providing a scaffolding role for the assembly of MBW (+ potentially others) proteins.

Response: Thank you for your careful work. We are sorry for our unclear description. We have revised the sentence here. Please find the newest one in the manuscript.

Line 69-71. The distinct of bHLH A and B is a new labelling system I have not seen before. I wonder why the authors haven't adopted bHLH1 (R/JAF13/Delila/EGL3) and bHLH2(IN/An1/Mut/TT8), as is used in many articles, including the model from Albert et al 2014 (Plant Cell), which this study is largely following.

Response: Thank you for your comment. Following your advice, we have changed bHLH A and B into bHLH1 and bHLH2 in current manuscript.

Line 73 Until (not Till)

Response: Thank you for your reminding. We have changed till into until here.

Line 84. MYBL2 belongs to the same clade of MYB repressor as PhMYB27, FaMYB1 etc. It is really unhelpful refer to it as an R3MYB without explanation or qualification. It has had a truncation of the first exon, which is why the R2 domain is largely missing. But it is phylogenetically and functionally related to the R2R3-MYB repressors, not the typical R3-MYBs. This matters because in the results, FaMYB4 is said to be closer in sequence to AtMYB4 (it is not). It is much closer to AtMYBL2.

Response: Thank you for your professional comment. We have added the explanation of AtMYBL2 in introduction and renamed FhMYB4 as FhMYB27.

Line 86 is missing Albert *et al* 2011 (Plant Journal) and 2014 (Plant Cell) are required for describing MYBx, MYB27 and the models being described.

Response: Thank you for your careful work. We have added the references here to describe the models. All the changes could be found in the newest manuscript.

Reviewer #2 (Remarks to the Author):

Two conclusions were made in this manuscript: the first one, FhMYB4 and FhCPC, belonged to R2R3-MYB and R3-MYB subgroup respectively, were isolated from Freesia flowers and functionally characterized. Functional studies indicated that they could suppress anthocyanin biosynthesis by inhibiting ABGs expression and might function in differential mechanisms according to repressor domains in the C-terminus; the second one is that two kinds of repressors, FhMYB4 and FhCPC, could also be activated by MBW complex with the purpose of balancing anthocyanin biosynthesis in Freesia. It is useful information to reveal the hierarchical and feedback anthocyanin regulation might be relatively conservative in angiosperm plants. However, there are still some recommends in the manuscript that need to be improved.

Response: Thank you for your summary and positive comments on our work. We are deeply honored for your recognition of our work.

1) For the methods section, lack of details of some experiments were found in the section, e.g. GUS activities measurement.

Response: Thank you for your valuable advice. We have added more details in the methods section. And the GUS activities measurement was also included in current version.

2) Detailed methods of statistics analysis used in the study were lost.

Response: Thank you for your careful reading. We have narrated the statistical analysis in figure legends. Moreover, we also added the "Statistics and reproducibility" section in the manuscript.

3) Some bootstrap values are relatively low, especially FhMYB4 and FhCPC highlighted with the red stars, which may lead to unconvincing clustering in fig. 1b. The NJ method for construction of the phylogenetic tree is not accurate and unacceptable, usually the ML tree (Maximum likelihood tree) may be a better choice. The gene trees built by different methods such as NJ, ML and MP methods,

have similar topological structure, so it can be considered reliable. Furthermore, an outgroup (from mosses or other lower plant species) should be added.

Response: Thank you for your professional comment. Following your advice, we re-built the phylogenetic tree by Maximum Likelihood method and the topological structure was similar like the earlier tree constructed by NJ method. Furthermore, the human c-MYB sequence (NP_001155129) was used for the outgroup, and a selection of non-flavonoid-related R2R3 MYBs from *Marchantia polymorpha* (MpMYB09 PTQ41991.1; MpMYB13 PTQ35332.1; MpMYB17 PTQ32714.1) were included for comparison.

4) There is a serious problem that the results of GD, GD-FhMYB4 and GD-FhCPC is inconsistency between Fig. 3b and Fig. 3c.

Response: Thank you for your careful reading of our manuscript. We are sorry for our unclear description. The data in Fig. 3b and Fig.3c was got from different systems. The purpose and principle are different. We have described the mechanisms and results in detail and added some references to help readers understand the results.

5) As far as I know, unpublished data cannot be used as a reference, but the author cites these documents more often.

Response: Thank you for your kind reminding. We are sorry for our frequent references to unpublished data. Fortunately, the manuscript about FhPAP1 has just been accepted and we have revised our manuscript in current version.

6) All species names require italics, such as *Arabidopsis thaliana*, *Marchantia*, *Freesia*, etc.

Response: Thank you for your careful work. We have revised our manuscript to assure the species names were all in italics.

7) The general rule, when the Latin name of species appears for the second time in the text, the generic name is usually abbreviated, for example, *A. thaliana*. There are many such writing errors in the text.

Response: Thank you for your kind reminding, the writing errors have been corrected.

8) Through a series of experiments, the author tries to prove that both novel MYB repressors that involved in the inhibition of anthocyanin synthesis in monocot is evolutionarily conservative as in dicot plant. However, more comparison between orthologous from dicot plant, for example, from *Arabidopsis*

should be supplied.

Response: Thanks for your valuable suggestions, orthologous in more eudicots such as *petunia*, *Arabidopsis*, *Mimulus*, poplar and grapevine were cited and compared in the current version.

9) More editing is needed to improve the organization of the text and English style, and to avoid inappropriate citations.

Response: Thank you for your advice. We also revised our manuscript carefully from beginning to end and asked a native English speaker to check the manuscript word by word to improve the English quality. Please find the newest version.

Reviewer #3 (Remarks to the Author):

Flower color pattern that resulted from anthocyanin is not only visually appealing, but also have implications in health and agriculture. Li, Shan and their colleagues reported two MYB anthocyanin repressors in monocot *Freesia hybrida*. One is an R2R3 MYB type, another is an R3 MYB type.

Through a series of painstaking experiments *in vivo* and *in vitro*, they convincingly demonstrated their two MYBs are true anthocyanin repressors. They found that these MYBs involved in anthocyanin regulation by interacting with the MBW complex. This MBW complex is also involved in a feedback regulatory loop, such as that reported in dicots. This work is a significant achievement for the authors, particularly these experiments are done in a non-model system.

However, there are a few things that I hope the authors can carry on to further improve this manuscript.

Response: On behalf of co-authors, we thank you very much for your recognition and positive comments on our work.

1 I disagree with the authors that the anthocyanin regulation is not well studied in monocot. Anthocyanin regulation is very well studied in several monocot crops. Therefore, I think it is an inaccurate description of the current understanding of the anthocyanin research.

Response: Thank you for your valuable advice. We are sorry for our inaccurate description. Anthocyanin regulation was also well studied in several monocots, such as maize, lily, orchid, onion and rice^{1-4,12-15}. However, few studies delved into the mutual regulation in monocots, especially in flowers, We have modified the inaccurate description in current manuscript.

2 Though the authors describe these two regulators and functionally validate them in *Arabidopsis* and tobacco, these transcription factors have already been well studied, including the mechanisms that the

authors worked out. In other words, even the authors have performed a lot of experiments, not much is really new if you take a broader look of the field. Many of these studies are largely confirmatory.

Response: Thank you for your comment. As you stated, the anthocyanin regulatory mechanism has been well studied and characterized to be conserved in dicotyledons. However, the regulatory mechanism was largely lagged in monocots. As review 1 stressed whether the MBW regulatory network models described for eudicots (Albert *et al.*, 2014, Plant Cell)¹¹ also exists in monocots was an interesting question, which until now has not been addressed because of the lack of a suitable model system (as raised in Albert *et al.*, 2014 Plant Signaling and Behavior)¹⁶. The grass models, such as rice and maize, are phylogenetically very distinct from other groups of monocots and lack the complex pigmentation patterning that is important for signaling to insect and animal pollinators. In addition, studies focusing on floral anthocyanin biosynthesis are beneficial to the molecular modification of horticultural plants. We hope to get your approval of our manuscript. Again, thank you for your suggestions on our manuscript.

3 The writing is poorly organized, many writings are not conventional. Please have some native speakers to check the writing. Throughout the manuscript, the authors start describing experiment before telling the purpose of the experiment, which makes it very hard to follow.

For example, the very first thing they did in the manuscript is to isolate the MYB orthologues. It is unclear to me why the authors want to study the MYB repressors in *Freesia hybrida*. Why understanding whether the anthocyanin regulation is conserved between monocot and dicot requires studying MYB repressors, there are lots of anthocyanin repressors other than MYBs. I also noticed the authors cited 27~37. It seems to me that the anthocyanin regulation in *Freesia hybrida* is similar to that in dicots, which would contradict author's claim in the introduction. All experiments are very well executed, but are not articulated to make it easier to understand. I also added a few comments in the attached manuscript and hope it helps.

Response: Thank you for your suggestion. We are sorry for our unclear description. We agree with you that anthocyanin biosynthetic pathway is conserved in plants. So the *Freesia* does. As for the references we cited, most of them were about structural genes in *F. hybrida*. It is speculative to conclude the anthocyanin regulation in *F. hybrida* is conserved compared to that in eudicots without robust data. As known, there are lots of anthocyanin regulators including activators and repressors have been reported in various plants. But it is unavoidable to mention the canonical MYB-bHLH-WD40 activation complex when we discuss anthocyanin regulation. Particularly, MYB factors in the complex might be the most specific and conspicuous regulators determining anthocyanin patterns. As one of the largest plant transcription factor families, MYB proteins have dual functions as direct activators or repressors. As we have characterized the components of MBW activation complex in *F. hybrida*⁶, it is interesting to further investigate whether the MBW regulatory network models described for eudicots (Albert *et al.*,

2014, Plant Cell)¹¹ also exists in monocots. As results, it seems to be largely conservative among angiosperm. As for the language, we are sorry for our unclear description. We have revised the manuscript thoroughly and asked a native English speaker to check the manuscript word by word to improve the English quality. All the changes could be found in current version which we hope could meet your requirement. Thanks for your comments in the attached manuscript and we tried our best to revise the manuscript following your suggestions.

4 Line 8 I would strongly disagree with this claim. The conservation of anthocyanin pathway has been well documented in several important monocot crops.

Response: Thank you for your comment. We are sorry for the unclear description and quite agree with you that the conservation of anthocyanin pathway has been well documented in several important monocot crops. We have modified the sentence to make it more accurate.

5. Line 9 Again, a large number of literatures on various of R2R3 MYB and R3 MYB repressors out there.

Response: Thank you for your work. We have revised the abstract, making it more accurate.

6. Line 12 “fine-tune”

Response: Thank you for your careful reading. We have revised the writing in the manuscript.

7. Line 52 Please add the *Mimulus* model

Response: Thank you for your valuable suggestion. The *Mimulus* model is a robust anthocyanin regulation system, we are sorry for the missing of the important references. We have modified the sentence following your advice, and *Mimulus* model was also mentioned in the Discussion section.

8. Line 73 “remains”

Response: Thank you for your careful work. We have corrected the mistakes.

9. Line 156 Reference about VP16 please

Response: Thank you for your constructive advice. We have added the reference here.

10. Line 241 Low resolution, can not really read any informative from this

Response: Thank you for your constructive advice. We have adjusted the characters and image resolution to make it readable.

11. Line 288 CPC in *Arabidopsis* and its orthologue in *Mimulus* are localized in nucleus and cytoplasm. What is the explanation for this discrepancy? If possible, I would also like to see the red channel to be displayed in Fig.S1 <https://dev.biologists.org/content/132/24/5387>
[https://www.cell.com/current-biology/pdfExtended/S0960-9822\(19\)31700-2](https://www.cell.com/current-biology/pdfExtended/S0960-9822(19)31700-2)

Response: Thank you for your valuable comment. We are sorry for our mistake. To validate our results, we have re-sequenced the vectors and re-constructed new constructs with GFP adding to the N or C termini of the repressors. The nuclear localization signal (NLS) - red fluorescent protein (RFP) was also included as a nuclear marker. The new image could be found in the latest version.

12. Line 301 What does the petal look like at different stages?

Response: Thank you for your constructive comment. We have added the *Freesia* pictures containing different developmental stages and tissues or organs. Moreover, the pictures and phenotypes can also be found in our earlier studies⁵⁻⁷.

13. Line 316 Not clear what effects do they have

Response: Thank you for your comment. We have modified the sentence here.

14. Line 322 ~~vividly~~

Response: Thank you for polishing the language. We have deleted the word in the sentence.

15. Line 579 These are experimental results supporting different repressive capability. But does not explain why they are different

Response: Thank you for careful reading. We are sorry for our unclear description. These are experimental results supporting different repressive capability as you said. We have modified the first sentence to make it clearer.

- 1 Hsu, C.-C., Chen, Y.-Y., Tsai, W.-C., Chen, W.-H. & Chen, H.-H. Three R2R3-MYB transcription factors regulate distinct floral pigmentation patterning in *Phalaenopsis* spp. *Plant physiology* **168**, 175-191, doi:10.1104/pp.114.254599 (2015).
- 2 Schwinn, K. E. *et al.* The Onion (*Allium cepa* L.) R2R3-MYB Gene MYB1 Regulates Anthocyanin Biosynthesis. *Frontiers in Plant Science* **7**, doi:10.3389/fpls.2016.01865 (2016).
- 3 Yamagishi, M., Shimoyamada, Y., Nakatsuka, T. & Masuda, K. Two R2R3-MYB Genes, Homologs of *Petunia* AN2, Regulate Anthocyanin Biosyntheses in Flower Tepals, Tepal Spots and Leaves of Asiatic Hybrid Lily. *Plant and Cell Physiology* **51**, 463-474, doi:10.1093/pcp/pcq011 (2010).
- 4 Yamagishi, M., Toda, S. & Tasaki, K. The novel allele of the LhMYB12 gene is involved in splatter-type spot formation on the flower tepals of Asiatic hybrid lilies (*Lilium* spp.). *New Phytologist* **201**, 1009-1020, doi:10.1111/nph.12572 (2014).
- 5 Li, Y. *et al.* Dihydroflavonol 4-Reductase Genes from *Freesia hybrida* Play Important and Partially Overlapping Roles in the Biosynthesis of Flavonoids. *Frontiers in Plant Science* **8**, doi:10.3389/fpls.2017.00428 (2017).
- 6 Li, Y. *et al.* The Conserved and Particular Roles of R2R3-MYB Regulator FhPAP1 from *Freesia hybrida* in Flower Anthocyanin Biosynthesis. *Plant and Cell Physiology*, doi:10.1093/pcp/pcaa065 (2020).
- 7 Shan, X. *et al.* A functional homologue of *Arabidopsis* TTG1 from *Freesia* interacts with bHLH proteins to regulate anthocyanin and proanthocyanidin biosynthesis in both *Freesia hybrida* and *Arabidopsis thaliana*. *Plant physiology and biochemistry : PPB* **141**, 60-72, doi:10.1016/j.plaphy.2019.05.015 (2019).
- 8 Meng, X. *et al.* Functional Differentiation of Duplicated Flavonoid 3-O-Glycosyltransferases in the Flavonol and Anthocyanin Biosynthesis of *Freesia hybrida*. *Frontiers in plant science* **10**, 1330-1330, doi:10.3389/fpls.2019.01330 (2019).
- 9 Li, Y. *et al.* The R2R3-MYB Factor FhMYB5 From *Freesia hybrida* Contributes to the

- Regulation of Anthocyanin and Proanthocyanidin Biosynthesis. *Frontiers in Plant Science* **9**, doi:10.3389/fpls.2018.01935 (2019).
- 10 Yang, Z. *et al.* MYB21 interacts with MYC2 to control the expression of terpene synthase genes in flowers of *Freesia hybrida* and *Arabidopsis thaliana*. *Journal of Experimental Botany*, doi:10.1093/jxb/eraa184 (2020).
- 11 Albert, N. W. *et al.* A Conserved Network of Transcriptional Activators and Repressors Regulates Anthocyanin Pigmentation in Eudicots. *Plant Cell* **26**, 962, doi:10.1105/tpc.113.122069 (2014).
- 12 Zheng, J. *et al.* Determining factors, regulation system, and domestication of anthocyanin biosynthesis in rice leaves. *New Phytologist* **223**, 705-721, doi:10.1111/nph.15807 (2019).
- 13 Anwar, M. *et al.* Ectopic Overexpression of a Novel R2R3-MYB, NtMYB2 from Chinese Narcissus Represses Anthocyanin Biosynthesis in Tobacco. *Molecules (Basel, Switzerland)* **23**, doi:10.3390/molecules23040781 (2018).
- 14 Anwar, M. *et al.* NtMYB3, an R2R3-MYB from Narcissus, Regulates Flavonoid Biosynthesis. *International journal of molecular sciences* **20**, doi:10.3390/ijms20215456 (2019).
- 15 Sakai, M., Yamagishi, M. & Matsuyama, K. Repression of anthocyanin biosynthesis by R3-MYB transcription factors in lily (*Lilium* spp.). *Plant Cell Reports* **38**, 609-622, doi:10.1007/s00299-019-02391-4 (2019).
- 16 Albert, N. W., Davies, K. M. & Schwinn, K. E. Gene regulation networks generate diverse pigmentation patterns in plants. *Plant Signal Behav* **9**, e29526-e29526, doi:10.4161/psb.29526 (2014).

Reviewers' comments:

Reviewer #1 (Remarks to the Author):

This revised manuscript by Li et al is a substantial improvement on the original submission, and I thank the authors for the care and attention taken to address the majority of the reviewers' comments and suggestions.

I do still have concerns about the sampling and replication. I asked for the nature of replication, and what each sample consisted of. The authors now state biological replicates, but there is no indication what they consisted of. What was the biological unit? Was pooling performed? The error bars (SD) are impossibly small for true biological replicates. I suspect a single pooled sample has been taken, and then three separate RNA isolations (for example) from the same pooled tissue sample. This is not a biological replicate.

It seems that this is a method being done quite widely to reduce error bars, but not appreciating that it is pseudoreplication and makes any statistical analysis completely invalid (to be clear, pooling can be used successfully but it must also be combined with biological replication - e.g. corolla tissue from 5 Stg 4 flowers from a single plant were used. Each replicate consisted of separate plants). It is up to the editor and journal to decide if they will accept studies without proper replication. I do consider the results look correct, but if pooled samples have been used without true biological replicates then the materials and methods must state this clearly and all statistical analyses removed.

Reviewer #2 (Remarks to the Author):

In the modified version, the authors addressed all my concerns. It is suggested that this manuscript is accepted by the journal of Communication Biology.

Reviewer #3 (Remarks to the Author):

The authors have addressed my concerns, however, there are a few things to note.

Line3, add "(MBW)" after "MYB-bHLH-WD40"

Line 7, insert "a" between in and non-grass

Line12, delete "one"

Line147, millennials? you mean million?

Line161, delete "MYB-bHLH-WD40"

Line163, supposed->hypothesized

Line214,horticultural ->horticultural

Line395->add "were" before "included"

Reviewers' comments:

Reviewer #1 (Remarks to the Author):

This revised manuscript by Li et al is a substantial improvement on the original submission, and I thank the authors for the care and attention taken to address the majority of the reviewers' comments and suggestions.

Response: Thanks for your affirmation of our revision, please receive our sincere appreciation to your help and kindness to review and improve our manuscript.

I do still have concerns about the sampling and replication. I asked for the nature of replication, and what each sample consisted of. The authors now state biological replicates, but there is no indication what they consisted of. What was the biological unit? Was pooling performed? The error bars (SD) are impossibly small for true biological replicates. I suspect a single pooled sample has been taken, and then three separate RNA isolations (for example) from the same pooled tissue sample. This is not a biological replicate.

It seems that this is a method being done quite widely to reduce error bars, but not appreciating that it is pseudoreplication and makes any statistical analysis completely invalid (to be clear, pooling can be used successfully but it must also be combined with biological replication - e.g. corolla tissue from 5 Stg 4 flowers from a single plant were used. Each replicate consisted of separate plants). It is up to the editor and journal to decide if they will accept studies without proper replication. I do consider the results look correct, but if pooled samples have been used without true biological replicates then the materials and methods must state this clearly and all statistical analyses removed.

Response: Thanks for your meticulous work and professional comments. We are sorry for the unclear description. We quite agree with you that it is pseudoreplication if taking

one single pooled tissue sample and then separating the RNA samples into several isolations for analysis. Actually, we collected flowers or tissues with highly uniform state of growth from several individuals to form one biological replicate, and three replicates were subjected to gene expression analysis. In detail, more than 5 flowers at each developmental stage from at least 3 individual plants were pooled to form one replicate. Floral tissues detached from Stg 5 flowers were also from more than 3 plants, and each replicate was pooled with more than 10 floral tissues. Total RNA from each replicate was extracted independently for further gene expression analysis, and three replicates were used in this study. Because each replicate was pooled from several highly uniformed flowers or tissues collected from plants cultivated at some conditions, the error bars (SD) might be smaller than those from routine sampling method. The sampling method has also been applied in our earlier studies¹⁻³.

As for the transient protoplast assay, enough quantities of protoplasts were prepared and divided into three replicates, each replicate of protoplasts were transfected with the same dose of plasmids and deemed as one biological replicate. In order to confirm the accuracy of the transient protoplast assay, protoplasts were isolated and prepared more than 3 times and the transfection were also repeated as mentioned above. Data in present manuscript were derived from the same batch with three biological repeats. The overall tendencies in each assay had been checked among batches of transfection and same tendencies were observed to prove the feasibility of the transfection system. The protoplast transfection experiment and sampling method we used in this study have also been widely used and accepted by many peers⁴⁻⁹. Following your advice, we have also added sufficient details and information in the latest version. We hope our latest manuscript could get your approval of publishing on *Communications Biology*, thank you for your reviewing of our manuscript and your professional and inspiring comments. If you have more concerns and suggestions, please let us know.

- 1 Shan, X. *et al.* The spatio-temporal biosynthesis of floral flavonols is controlled by differential phylogenetic MYB regulators. *New phytologist*, minor review (2020).
- 2 Yang, Z. *et al.* MYB21 interacts with MYC2 to control the expression of terpene synthase genes in flowers of *Freesia hybrida* and *Arabidopsis thaliana*. *Journal of Experimental Botany*, doi:10.1093/jxb/eraa184 (2020).
- 3 Gao, F. *et al.* Identification and characterization of terpene synthase genes accounting for volatile terpene emissions in flowers of *Freesia x hybrida*. *Journal of Experimental Botany* **69**, 4249-4265, doi:10.1093/jxb/ery224 (2018).
- 4 Tiwari, S. B., Hagen, G. & Guilfoyle, T. The Roles of Auxin Response Factor Domains in Auxin-Responsive Transcription. *Plant Cell* **15**, 533-543, doi:10.1105/tpc.008417 (2003).
- 5 Wang, S., Tiwari, S. B., Hagen, G. & Guilfoyle, T. J. AUXIN RESPONSE FACTOR7 restores the expression of auxin-responsive genes in mutant *Arabidopsis* leaf mesophyll protoplasts. *Plant Cell* **17**, 1979-1993, doi:10.1105/tpc.105.031096 (2005).
- 6 Tian, H. *et al.* NTL8 Regulates Trichome Formation in *Arabidopsis* by Directly Activating R3 MYB Genes TRY and TCL1. *Plant Physiol* **174**, 2363-2375, doi:10.1104/pp.17.00510 (2017).
- 7 Dai, X. *et al.* A single amino acid substitution in the R3 domain of GLABRA1 leads to inhibition of trichome formation in *Arabidopsis* without affecting its interaction with GLABRA3. *Plant, Cell & Environment* **39**, 897-907, doi:10.1111/pce.12695 (2016).

- 8 Zheng, K. *et al.* Involvement of PACLOBUTRAZOL RESISTANCE6/KIDARI, an Atypical bHLH Transcription Factor, in Auxin Responses in Arabidopsis. *Frontiers in Plant Science* **8**, doi:10.3389/fpls.2017.01813 (2017).
- 9 Shin, R. *et al.* The Arabidopsis transcription factor MYB77 modulates auxin signal transduction. *Plant Cell* **19**, 2440-2453, doi:10.1105/tpc.107.050963 (2007).

Reviewer #2 (Remarks to the Author):

In the modified version, the authors addressed all my concerns. It is suggested that this manuscript is accepted by the journal of *Communication Biology*.

Response: We do appreciate for your approval of our manuscript. On behalf of co-authors, we thank you very much for your positive comments and suggestions on our manuscript.

Reviewer #3 (Remarks to the Author):

The authors have addressed my concerns, however, there are a few things to note.

Line3, add "(MBW)" after "MYB-bHLH-WD40"

Response: Thank you for your revise. We have added "(MBW)" after "MYB-bHLH-WD40".

Line 7, insert "a" between in and non-grass

Response: Thank you for your careful reading. We have inserted "a" between “in” and “non-grass”.

Line12, delete "one"

Response: Thank you for your meticulous work. We have deleted "one" following your advice.

Line147, millennials? you mean million?

Response: Thank you for your comment. We have changed “millennials” into “million years”.

Line161, delete "MYB-bHLH-WD40"

Response: Thank you for your careful work. We have deleted "MYB-bHLH-WD40" here.

Line163, supposed->hypothesized

Response: Thank you for your polishing the language. We have changed “supposed” into "hypothesized" here.

Line214, horticultural ->horticultural

Response: We are sorry for the carelessness. We have revised the word here.

Line395->add "were" before "included"

Response: Thank you for your careful reading. We have added "were" before "included" following your advice. Moreover, we really enjoyed the great benefit of your instructions. On behalf of co-authors, we thank you very much for your positive comments and suggestions on our manuscript.